# Scalable Fair Influence Maximization

**Xiaobin Rui**
China University of Mining and Technology
Xuzhou, Jiangsu, China
`ruixiaobin@cumt.edu.cn`

**Zhixiao Wang**[*]
China University of Mining and Technology
Xuzhou, Jiangsu, China
`zhxwang@cumt.edu.cn`

**Jiayu Zhao**
China University of Mining and Technology
Xuzhou, Jiangsu, China
`zhaojy@cumt.edu.cn`

**Lichao Sun**
Lehigh University
Bethlehem, PA, USA
`lis221@lehigh.edu`

**Wei Chen**[*]
Microsoft Research Asia
Beijing, China,
`weic@microsoft.com`

## Abstract

Given a graph $G$, a community structure $\mathcal{C}$, and a budget $k$, the fair influence maximization problem aims to select a seed set $S$ ($|S| \leq k$) that maximizes the influence spread while narrowing the influence gap between different communities. While various fairness notions exist, the welfare fairness notion, which balances fairness level and influence spread, has shown promising effectiveness. However, the lack of efficient algorithms for optimizing the welfare fairness objective function restricts its application to small-scale networks with only a few hundred nodes. In this paper, we adopt the objective function of welfare fairness to maximize the exponentially weighted summation over the influenced fraction of all communities. We first introduce an unbiased estimator for the fractional power of the arithmetic mean. Then, by adapting the reverse influence sampling (RIS) approach, we convert the optimization problem to a weighted maximum coverage problem. We also analyze the number of reverse reachable sets needed to approximate the fair influence at a high probability. Further, we present an efficient algorithm that guarantees $1 - 1/e - \varepsilon$ approximation.

## 1   Introduction

Influence maximization (IM) is a well-studied problem in the field of social network analysis. Given a graph $G$ and a positive integer $k$, the problem asks to find a node set $S$ ($|S| \leq k$) which can spread certain information to trigger the largest expected number of remaining nodes. There have been various IM variants, such as adaptive influence maximization [1], multi-round influence maximization [2], competitive influence maximization [3], and time-critical influence maximization [4]. Influence maximization and these variants have important applications in viral marketing, rumor control, health interventions, etc [5].

Considering the situation that when disseminating public health interventions, for example, suicide/HIV prevention [6] or community preparedness against natural disasters, we can select individuals who act as peer-leaders to spread such information to maximize the outreach following influence

---

[*]Corresponding author

37th Conference on Neural Information Processing Systems (NeurIPS 2023).

maximization. However, it may lead to discriminatory solutions as individuals from racial minorities or LGBTQ communities may be disproportionately excluded from the benefits of the intervention [7]. Therefore, derived from such significant social scenarios, fairness in influence maximization has become a focus of attention for many researchers [8, 9, 7, 10].

Generally, fair influence maximization aims to improve the influenced fraction inside some communities where the coverage may get unfairly low. Currently, a universally accepted definition of fair influence maximization remains elusive, and recent work has incorporated fairness into influence maximization by proposing various notions of fairness, such as maximin fairness [9], diversity constraints [8], demographic parity [11], and welfare fairness [7]. Among these notions, welfare fairness show several attractive features. Its objective function is the weighted sum (with community size as the weight) of the fractional power ($\alpha$ fraction) of the expected proportion of activated nodes within every community. The fractional exponent $\alpha$ is the inequality aversion parameter, allowing one to balance between fairness and influence spread, with $\alpha$ tending to 1 for influence spread and $\alpha$ tending to 0 for fairness. The objective function enjoys both monotonicity and submodularity, enabling a greedy approach with $1 - 1/e - \varepsilon$ approximation.

However, even though Rahmattalabi *et al.* [9] has given a full analysis of welfare fairness, there is currently no efficient algorithm to optimize its objective function with provable guarantee, which restricts their applications to small-scale networks with only a few hundred nodes. In this paper, we propose an efficient algorithm for maximizing the welfare fairness based on reverse influence sampling (RIS) [12, 13, 14]. The main challenges in adapting the RIS approach to the welfare fairness objective include: (a) how to carry out the unbiased estimation of the fractional power of the expected proportion of activated nodes in each community, since simply obtaining an unbiased estimate of the expected proportion and then taking its fractional power is not an unbiased estimator; (b) how to integrate the unbiased estimation designed into the RIS framework. We address both challenges and propose a new scalable fair influence maximization with theoretical guarantees.

Our contributions can be summarized as follows:

- We propose an unbiased estimator for the fractional power of the arithmetic mean by leveraging Taylor expansion techniques. The estimator enables us to accurately estimate the fair influence under welfare fairness.

- Based on the above unbiased estimator, we adapt the RIS approach to approximate the fair influence with Reverse Reachable (RR) sets and propose the FIMM algorithm that works efficiently while guaranteeing the $(1 - 1/e - \varepsilon)$-approximation solution. Our theoretical analysis needs to address the concentration of the unbiased estimator of the fractional power and thus is much more involved than the standard RIS analysis.

- We carry out detailed experimental analysis on five real social networks to study the trade-off between fairness and total influence spread. We test different fairness parameters, influence probabilities, seed budget, and community structures to confirm the performance of our proposed algorithms.

**Related Work**    Influence maximization (IM) is first studied as an algorithmic problem by Domingos and Richardson [15, 16]. Kempe *et al.* [17] mathematically formulate IM as a discrete optimization problem and prove it is NP-hard. They also provide a greedy algorithm with $1 - 1/e$ approximation based on the submodularity and monotonicity of the problem. Hence, many works have been proposed to improve the efficiency and scalability of influence maximization algorithms [18, 19, 20, 21, 22, 12, 13, 14]. Among these methods, the most recent and the state of the art is the reverse influence sampling (RIS) approach [12, 13, 14, 23], where the IMM algorithm [14] is one of the representative algorithms. The idea of RIS approaches is to generate an adequate number of reverse reachable sets (*a.k.a.* RR sets), and then the influence spread can be approximated at a high probability based on these RR sets. Therefore, the greedy approach can be easily applied by iteratively selecting the node that could bring the maximal marginal gain in terms of influence spread as a seed node.

Recent work has incorporated fairness directly into the influence maximization framework by relying on Rawlsian theory [9], game theoretic principles [8], and equity-based notion [24]. Based on the Rawlsian theory, maximin fairness [9, 8] aims to maximize the influence fraction of the worst-off community. Inspired by the game theoretic notion of core, diversity constraints [8] require that every community obtains an influenced fraction higher than when it receives resources proportional to its size and allocates them internally. Equity-based notion [24] strives for equal influenced fraction

across all communities. However, these notions can hardly balance fairness and total influence and usually lead to a high influence reduction. Especially, strict equity [24] is rather hard to achieve in influence maximization. To address these shortcomings, Rahmattalabi *et al.* [7] propose the welfare fairness that can control the trade-off between fairness and total influence by an inequality aversion parameter. Based on the cardinal welfare theory [25], the objective function of welfare fairness is to maximize the weighted summation over the exponential influenced fraction of all communities. Fish *et al.* [26] also follow welfare functions and propose $\phi$-mean fairness, where the objective function becomes MMF when $\phi = -\infty$. However, they do not address the challenge of unbiased estimation of the fractional power. In addition, none of the above studies address the scalability of the algorithms. Thus, to the best of our knowledge, we are the first to study scalability in the context of fair influence maximization.

## 2 Model and Problem Definition

**Information Diffusion Model**    In this paper, we adopt the well-studied *Independent Cascade (IC) model* as the basic information diffusion model. Under IC model, a social network is modeled as a directed influence graph $G = (V, E, p)$, where $V$ is the set of vertices (nodes) and $E \subseteq V \times V$ is the set of directed edges that connect pairs of nodes. For an edge $(v_i, v_j) \in E$, $p(v_i, v_j)$ indicates the probability that $v_i$ influences $v_j$. The diffusion of information or influence proceeds in discrete time steps. At time $t = 0$, the *seed set* $S$ is selected to be active, denoted as $A_0$. At each time $t \geq 1$, all nodes in $A_{t-1}$ try to influence their inactive neighbors following influence probability $p$. The set of activated nodes at step $t$ is denoted as $A_t$. The diffusion process ends when there is no more node activated in a time step. An important metric in influence maximization is the *influence spread*, denoted as $\sigma(S)$, which is defined as the expected number of active nodes when the propagation from the given seed set $S$ ends. For the IC model, $\sigma(S) = \mathbb{E}[|A_0 \cup A_1 \cup A_2 \cup \dots|]$. We use $ap(v, S)$ to represent the probability that node $v$ is activated given the seed set $S$. Then we have $\sigma(S) = \sum_{v \in V} ap(v, S)$.

**Live-edge Graph**    Given the influence probability $p$, we can construct the live-edge graph $L = (V, E(L))$, where each edge $(v_i, v_j)$ is selected independently to be a *live edge* with the probability $p(v_i, v_j)$. The influence diffusion in the IC model is equivalent to the deterministic propagation via bread-first traversal in a random live-edge graph $L$. Let $\Gamma(G, S)$ denote the set of nodes in graph $G$ that can be reached from the node set $S$. By the above live-edge graph model, we have $\sigma(S) = \mathbb{E}_L[|\Gamma(L, S)|] = \sum_L Pr[L|G] \cdot |\Gamma(L, S)|$, where the expectation is taken over the distribution of live-edge graphs, and $Pr[L|G]$ is the probability of sampling a live-edge graph $L$ in graph $G$.

**Approximation Solution**    A set function $f : V \to \mathbb{R}$ is called *submodular* if for all $S \subseteq T \subseteq V$ and $u \in V \setminus T$, $f(S \cup \{u\}) - f(S) \geq f(T \cup \{u\}) - f(T)$. Intuitively, submodularity characterizes the diminishing return property often occurring in economics and operation research. Moreover, a set function $f$ is called *monotone* if for all $S \subseteq T \subseteq V$, $f(S) \leq f(T)$. It is shown in [17] that influence spread $\sigma(\cdot)$ for the independent cascade model is a monotone submodular function. A non-negative monotone submodular function allows a greedy solution to its maximization problem with $1 - 1/e$ approximation [27], which provides the technical foundation for most influence maximization tasks.

**Fair Influence Maximization**    For a given graph $G$ with $n_G$ nodes, the classic influence maximization problem is to choose a seed set $S$ consisting of at most $k$ seeds to maximize the influence spread $\sigma(S, G)$. Assuming each node belongs to one of the disjoint communities $c \in \mathcal{C} := \{1, 2, \dots, C\}$, such that $V_1 \cup V_2 \cup \dots \cup V_C = V$ where $V_c$ ($n_c = |V_c|$) denotes the set of nodes that belongs to community $c$. Generally, *fair influence maximization (FIM)* aims to narrow the influence gap between different communities while maintaining the total influence spread as much as possible. In this paper, we adopt the fair notion proposed by Rahmattalabi *et al.* [7], where the welfare function is used to aggregate the cardinal utilities of different communities. The goal is to select at most $k$ seed nodes, such that the objective function $F_\alpha(S)$ (also referred to as fair influence in this paper) is maximized, where $F_\alpha(S) = \sum_{c \in \mathcal{C}} n_c \boldsymbol{u}_c(S)^\alpha, 0 < \alpha < 1$. The utility $\boldsymbol{u}_c(S)$ denotes the expected proportion of influenced nodes in the community $c$ with the seed set $S$. Exponent $\alpha$ is the inequality aversion parameter that controls the trade-off between fairness and total influence, with $\alpha$ tending to 1 for influence spread and $\alpha$ tending to 0 for fairness. We thus define the fair influence maximization problem in this paper as follows:

**Definition 1.** *The Fair Influence Maximization (FIM) under the independent cascade model is the optimization task where the input includes the directed influence graph $G = (V, E, p)$, the non-overlapping community structure $\mathcal{C}$, and the budget $k$. The goal is to find a seed set $S^*$ to maximize the fair influence, i.e., $S^* = \text{argmax}_{S:|S| \leq k} F_\alpha(S)$.*

According to [7], the fair influence $F_\alpha(S)$ is both monotone and submodular, which provides the theoretical basis for our efficient algorithm design, to be presented in the next section.

## 3   Method

The monotonicity and submodularity of the fair influence objective function enable a greedy approach for the maximization task. However, as commonly reported in influence maximization studies, naively implementing a greedy approach directly on the objective function will suffer a long running time. The reason is that accurate function evaluation requires a large number of Monte-Carlo simulations. In this section, we aim to significantly speed up the greedy approach by adapting the reverse influence sampling (RIS) [12, 13, 14], which provides both theoretical guarantee and high efficiency. We propose the FIMM algorithm that is efficient when the number of communities is small, which is hopefully a common situation such as gender and ethnicity.

For the convenience of reading, we list most important symbols featured in this paper in Table 1.

Table 1: Important symbols appeared in this paper.

| Symbol | Explanation |
|---|---|
| $G = (V, E, p)$ | A network; |
| $V$ | Node set of the network; |
| $E$ | Edge set of the network; |
| $n_G$ | The number of nodes in $G$, i.e. $n_G = |V|$; |
| $p(v_i, v_j)$ | The probability that $v_i$ influence $v_j$; |
| $\mathcal{C} = \{c_1, c_2, \cdots\}$ | Community structure; |
| $C$ | The number of communities in $\mathcal{C}$; |
| $V_c$ | The node set in community $c$; |
| $n_c$ | The number of nodes in community $c$, i.e. $n_c = |V_c|$; |
| $S$ | A seed set; |
| $S^*$ | The optimal seed set for fair influence maximization; |
| $ap(v, S)$ | The expected probability that $v$ is activated by $S$; |
| $\sigma(S)$ | Influence spread of $S$, i.e. $\sigma(S) = \sum_{v \in V} ap(v, S)$; |
| $\boldsymbol{u}_c$ | The utility of $c$ (expected fraction of influenced nodes in $c$); |
| $F_\alpha(S)$ | The fair influence of $S$; |
| $\mathcal{R}$ | A set of RR sets; |
| $\mathcal{R}_c$ | The set of RR sets rooted in community $c$; |
| $\hat{F}_\alpha(S, \mathcal{R})$ | The unbiased estimator for fair influence of $S$ based on $\mathcal{R}$; |
| $\theta$ | The total number of RR sets; |
| $\theta_c$ | The number of RR sets rooted in community $c$; |
| $\alpha$ | The aversion parameter regarding fairness; |
| $Q$ | The approximation parameter for Taylor expansion; |
| $\varepsilon$ | The accuracy parameter; |
| $\ell$ | The confidence parameter. |

### 3.1   Unbiased Fair Influence

To estimate the influence spread, we may generate a number of live-edge graphs $\mathcal{L} = \{L_1, L_2, \cdots, L_t\}$ as samples. Then, for a given seed set $S$, $\hat{\sigma}(\mathcal{L}, S) = \frac{1}{t} \sum_{i=1}^{t} |\Gamma(L_i, S)|$ is an unbiased estimator of $\sigma(S)$. However, situations are completely different for fair influence. For each community $c$, its fair influence is actually $n_c \boldsymbol{u}_c^\alpha$. If we still generate a number of live-edge graphs and estimate $\boldsymbol{u}_c$ by $\hat{\boldsymbol{u}}_c(\mathcal{L}, S) = \frac{1}{t} \sum_{i=1}^{t} |\Gamma(L_i, S) \cap V_c| / |V_c|$, then $\hat{\boldsymbol{u}}_c(\mathcal{L}, S)$ is an unbiased estimator for $\boldsymbol{u}_c$, but $\hat{\boldsymbol{u}}_c(\mathcal{L}, S)^\alpha$ is actually a biased estimator of $\boldsymbol{u}_c^\alpha$ for $0 < \alpha < 1$. In fact, the value of $\boldsymbol{u}_c^\alpha$ is generally higher than the true value, which is revealed by Jensen's Inequality.

**Fact 1.** *(Jensen's Inequality) If $X$ is a random variable and $\phi$ is a concave function, then*

$$\mathbb{E}[\phi(X)] \leq \phi(\mathbb{E}[X]).$$

Therefore, our first challenge in dealing with the welfare fairness objective is to provide an unbiased estimator for the fractional power value of $\boldsymbol{u}_c^\alpha$. We meet this challenge by incorporating Taylor expansion as in Lemma 1.

**Lemma 1.** *For a given seed set $S$ and an inequality aversion parameter $\alpha$, the fair influence*

$$F_\alpha(S) = \sum_{c \in \mathcal{C}} n_c \left(1 - \alpha \sum_{n=1}^\infty \eta(n, \alpha)\big(1 - \boldsymbol{u}_c(S)\big)^n\right), \eta(n, \alpha) = \begin{cases} 1, & n = 1, \\ \frac{(1-\alpha)(2-\alpha)...(n-1-\alpha)}{n!}, & n \geq 2. \end{cases}$$

*Proof.* By Taylor expansion of binomial series, we have

$$(1 + x)^\alpha = 1 + \sum_{n=1}^\infty \binom{\alpha}{n} x^n, \binom{\alpha}{n} = \frac{\alpha(\alpha - 1)...(\alpha - n + 1)}{n!}.$$

By definition of fair influence in Definition 1, we have

$$F_\alpha(S) = \sum_{c \in \mathcal{C}} n_c \big(1 + \big(\boldsymbol{u}_c(S) - 1\big)\big)^\alpha = \sum_{c \in \mathcal{C}} n_c \left(1 + \sum_{n=1}^\infty \binom{\alpha}{n}\big(\boldsymbol{u}_c(S) - 1\big)^n\right)$$

$$= \sum_{c \in \mathcal{C}} n_c \left(1 - \alpha \sum_{n=1}^\infty \eta(n, \alpha)\big(1 - \boldsymbol{u}_c(S)\big)^n\right) \quad (1)$$

where

$$\eta(n, \alpha) = \begin{cases} 1, & n = 1, \\ \frac{(1-\alpha)(2-\alpha)...(n-1-\alpha)}{n!}, & n \geq 2. \end{cases}$$

Thus concludes the proof. $\qquad\qquad\square$

Lemma 1 demonstrates that the calculation of fair influence with fractional powers can be transformed into the summation of integral powers. Further, we can get an unbiased estimator for integral powers of arithmetic mean as given in Lemma 2.

**Lemma 2.** *[28] Suppose that a simple random sample of size $m$ is to be drawn, with replacement, in order to estimate $\mu^n$. An unbiased estimator for $\mu^n$ ($n \leq m$) is*

$$\hat{\mu}^n = \frac{(m - n)!}{m!}\{\sum x_{i_1} x_{i_2} \cdots x_{i_n}\}(i_1 \neq i_2 \neq \cdots \neq i_n) \quad (2)$$

*where the summation extends over all permutations of all sets of $n$ observations in a sample subject only to the restriction noted.*

### 3.2 Unbiased Fair Influence with RR sets

Many efficient influence maximization algorithms such as IMM [14] are based on the RIS approach, which generates a suitable number of reverse-reachable (RR) sets for influence estimation.

**RR set**  An RR set $RR(v)$ (rooted at node $v \in V$) can be generated by reversely simulating the influence diffusion process starting from the root $v$, and then adding all nodes reached by reversed simulation into this RR set. In this way, $RR(v)$ is equivalent to collecting all nodes that can reach $v$ in the random live-edge graph $L$, denoted by $\Gamma'(L, v)$. Intuitively, each node $u \in RR(v)$ if selected as a seed would activate $v$ in this random diffusion instance. We say that $S$ covers a RR set $RR(v)$ if $S \cap RR(v) \neq \varnothing$. The expected activated probability $ap(v, S)$ is thus equivalent to the probability that $S$ covers a randomly generated $v$-rooted RR set. In the following, we use $RR(v)$ to represent a randomly generated RR set when $v$ is not specified, *i.e.*, $RR(v) = \Gamma'_{L \sim \mathcal{U}(P_L)}(L, v)$ where $P_L$ is the space of all live-edge graphs and $\mathcal{U}(\cdot)$ denotes the uniform distribution.

Let $X_c$ be the random event that indicates whether a randomly selected node in community $c$ would be influenced in a diffusion instance by the given seed set $S$. As mentioned above, an RR set maps to a random diffusion instance. Assuming we generate $\mathcal{R}$ consisting of $\theta$ RR sets in total and each community $c$ gets $\theta_c$ RR sets. Let $\mathcal{R}_c$ be the set of RR sets that are rooted in the community $c$, then $|\mathcal{R}_c| = \theta_c$. Let $X_c^i$ ($i \in [\theta_c]$) be a random variable for each RR set $R_i \in \mathcal{R}_c$, such that $X_c^i = 1$ if $\mathcal{R}_c^i \cap S \neq \varnothing$, and $X_c^i = 0$ otherwise. Then, we have $\mathbb{E}[X_c] = \boldsymbol{u}_c$ and $\mathbb{E}[\overline{X_c}] = 1 - \boldsymbol{u}_c$.

Based on Lemma 1 and Lemma 2, we can get the unbiased estimator of $\mathbb{E}[X_c]^\alpha$ through RR sets as

$$
\begin{aligned}
\mathbb{E}[X_c]^\alpha &= 1 - \alpha \sum_{n=1}^{\infty} \eta(n, \alpha)(1 - \mathbb{E}[X_c])^n \\
&= 1 - \alpha \sum_{n=1}^{\infty} \eta(n, \alpha)\frac{(\theta_c - n)!}{\theta_c!}\left\{\sum \overline{X_c^{i_1}} \cdot \overline{X_c^{i_2}} \cdots \overline{X_c^{i_n}}\right\},
\end{aligned}
\tag{3}
$$

Further, we can get the unbiased estimator of the fair influence $F_\alpha(S)$ as

$$
\begin{aligned}
\hat{F}_\alpha(S, \mathcal{R}) &= \sum_{c \in \mathcal{C}} n_c \left(1 - \alpha \sum_{n=1}^{\infty} \eta(n, \alpha)\frac{(\theta_c - n)!}{\theta_c!}\left\{\sum \overline{X_c^{i_1}} \cdot \overline{X_c^{i_2}} \cdots \overline{X_c^{i_n}}\right\}\right) \\
&= \sum_{c \in \mathcal{C}} n_c \left(1 - \alpha \sum_{n=1}^{\infty} \eta(n, \alpha)\prod_{i=0}^{n-1}\frac{\pi_c - i}{\theta_c - i}\right)
\end{aligned}
\tag{4}
$$

where $\pi_c = \theta_c - \sum_{i \in [\theta_c]} X_c^i$, and

$$
\eta(n, \alpha) = \begin{cases} 1, & n = 1, \\ \frac{(1-\alpha)(2-\alpha)\ldots(n-1-\alpha)}{n!}, & n \geq 2. \end{cases}
$$

In the following, we consider Eq. 4 as our objective function to deal with the fair IM problem.

## 3.3 FIMM

For a given seed set $S$, let $\varphi[c]$ denote the number of all $u$-rooted ($u \in V_c$) RR sets covered by $S$, and $\kappa[v][c]$ denote the number of all $u$-rooted ($u \in V_c$) RR sets that covered by $v$ ($v \in V \setminus S$) but not by $S$, then the marginal fair influence gain of $v$ is

$$
\begin{aligned}
\hat{F}_\alpha(v|S) &= \sum_{c \in \mathcal{C}} n_c \left(1 - \alpha \sum_{n=1}^{\infty} \eta(n, \alpha)\prod_{i=0}^{n-1}\frac{\theta_c - \kappa[v][c] - \varphi[c] - i}{\theta_c - i}\right) \\
&\quad - \sum_{c \in \mathcal{C}} n_c \left(1 - \alpha \sum_{n=1}^{\infty} \eta(n, \alpha)\prod_{i=0}^{n-1}\frac{\theta_c - \varphi[c] - i}{\theta_c - i}\right) \\
&= \sum_{c \in \mathcal{C}} \alpha n_c \sum_{n=1}^{\infty} \eta(n, \alpha)\left(\prod_{i=0}^{n-1}\frac{\theta_c - \varphi[c] - i}{\theta_c - i} - \prod_{i=0}^{n-1}\frac{\theta_c - \kappa[v][c] - \varphi[c] - i}{\theta_c - i}\right)
\end{aligned}
\tag{5}
$$

Therefore, when generating RR sets, we have to count $\kappa[v][c]$ which indicates the community-wise coverage for $v$ and record $\eta[v]$ which indicates the linked-list from $v$ to all its covered RR sets, as shown in Algorithm 1. As shown in lines 6~9, when generating a random $v$-rooted RR set $RR(v)$, we count all nodes $u \in RR(v)$ and raise all $\kappa[u][c(v)]$ by 1, where $c(v)$ indicates $v$'s community label. It should be noted that modifying $\kappa[u][c(v)]$ can be accomplished simultaneously when generating $RR(v)$ by the reverse influence sampling.

Based on the RR sets generated by Algorithm 1, we present our FIMM algorithm (Algorithm 2) to select $k$ seed nodes that maximize Eq. 4 through a greedy approach, *i.e.*, iteratively selecting a node with the maximum alternative marginal fair influence gain as presented in Eq. 5. Apparently, it costs $O(C)$ to calculate $\hat{F}(v|S)$ for any $v$ where $C$ is the number of communities. When $C$ is small (*i.e.*, a constant), it would be efficient to compute $\hat{F}(v|S)$ for all $v \in V$ in $O(Cn_G)$. Besides, since

**Algorithm 1:** RR-Generate: Generate RR sets

---

**Input:** Graph $G = (V, E, p)$, community $\mathcal{C}$, budget $k$, number of RR sets for each community $\theta_c$
**Output:** RR sets $\mathcal{R}$, community-wise coverage $\kappa$, linked-list $\eta$ from nodes to covered RR sets

1 Initialize $\kappa[v][c] = 0$ for all $v \in V$, $c \in \mathcal{C}$;
2 Initialize $\eta[v] = \varnothing$ for all $v \in V$;
3 $\mathcal{R} = \varnothing$;
4 **for** $c \in \mathcal{C}$ **do**
5     **for** $i = 1$ *to* $\theta_c$ **do**
6         Select a random node $v$ in community $c$;
7         Sample a random RR set $R = RR(v)$;
8         **for** $u \in R$ **do**
9             $\kappa[u][c(v)] = \kappa[u][c(v)] + 1$;
10         $\mathcal{R} = \mathcal{R} \cup \{R\}$;
11         $\eta[v] = \eta[v] \cup \{R\}$;

---

$\hat{F}_\alpha(S, \mathcal{R})$ is submodular and monotone, we can adopt a lazy-update strategy [18] that selects $v$ with the maximal $\hat{F}_\alpha(v|S)$ as a seed node if $\hat{F}_\alpha(v|S)$ is still the maximum after updating. This lazy-update strategy (lines 10∼12) can cut down a great amount of redundant time cost that can be empirically up to 700 times faster than a simple greedy algorithm [18].

There are two vital counting arrays in Algorithm 2, *i.e.*, $\varphi[c]$ and $\kappa[v][c]$. $\varphi[c]$ records and updates the number of RR sets covered by $S$ in community-wise. By lines 20∼24, $\kappa[v][c]$ keeps updating and always indicates the extra coverage of $v$ on all $u$-rooted ($u \in V_c$) RR sets besides $S$. It establishes a

---

**Algorithm 2:** FIMM: Fair Influence Maximization

---

**Input:** Graph $G = (V, E, p)$, community $\mathcal{C}$, budget $k$, approximation parameter $Q$
**Output:** Seed set $S$

1 $(\mathcal{R}, \kappa, \eta) = $ RR-Generate$(G, \mathcal{C}, k, \theta_c)$
2 Initialize $\varphi[c] = 0$ for all $c \in \mathcal{C}$; //indicating the number of covered RR sets rooted in $c$
3 Initialize $\gamma(v)$ according to Eq.(5) for all $v \in V$; //indicating initial marginal gain
4 Initialize $covered[R] = false$ for all $R \in \mathcal{R}$; //indicating whether $R$ is covered
5 Initialize $updated[v] = true$ for all $v \in V$; //indicating whether $\kappa[v]$ is updated
6 $S = \varnothing$;
7 **for** $i = 1$ *to* $k$ **do**
8     **while** *true* **do**
9         $v = \arg\max_{u \in V \setminus S} \gamma(u)$;
10         **if** $updated(v) == false$ **then**
11             Updating $\gamma(v)$ according to Eq.(5);
12             $updated(v) = true$;
13         **else**
14             $S = S \cup \{v\}$;
15             **for** $v \in V$ **do**
16                 $updated(v) = false$;
17             **break**;
18     **for** $c \in \mathcal{C}$ **do**
19         $\varphi[c] = \varphi[c] + \kappa[v][c]$;
20     **for** *all* $R \in \eta[v] \wedge covered[R] == false$ **do**
21         $covered[R] = true$;
22         $r = root(R)$;
23         **for** *all* $u \in R \wedge u \neq v$ **do**
24             $\kappa[u][c(r)] = \kappa[u][c(r)] - 1$;

---

convenient way for updating $\varphi(c)$ that only needs to increase $\varphi(c)$ by $\kappa[v][c]$ where $v$ is the newly selected node for all $c \in \mathcal{C}$. If we denote the original community-wise coverage as $\kappa'$, which means $(\sim, \kappa', \sim) = \mathsf{RR\text{-}Generate}(G, \mathcal{C}, k, \theta_c)$, then it holds $\kappa'[v][c] = \kappa[v][c] + \varphi[c]$ for all $v \in V$ and $c \in \mathcal{C}$ in Algorithm 2.

### 3.4 Number of RR sets

In this subsection, we discuss the number of RR sets needed to approximate the fair influence with high probability. Let $OPT_F$ denote the optimal solution and $S^*$ denote the corresponding optimal seed set for the fair influence maximization problem defined in this paper, *i.e.*, $OPT_F = F_\alpha(S^*) = \sum_{c \in \mathcal{C}} n_c \boldsymbol{u}_c(S^*)^\alpha = \sum_{c \in \mathcal{C}} n_c \left(1 - \alpha \sum_{n=1}^\infty \eta(n, \alpha)\left(1 - \boldsymbol{u}_c(S^*)\right)^n\right)$. Since this paper deals with the fair influence maximization problem, we thus assume that the maximal community utility $max_{c \in \mathcal{C}} \boldsymbol{u}_c(S^\#) \geq max_{c \in \mathcal{C}} \boldsymbol{u}_c(S^*)$ of an arbitrary seed set $S^\#$ would not be too big.

**Lemma 3.** *Let* $\delta_1 \in (0, 1)$, $\varepsilon_1 \in (0, 1)$, *and* $\theta_1 = \frac{12Q^2 \ln(C/\delta_1)}{\varepsilon_1^2(1-b)}$ *where $Q$ is the approximation parameter,* $b = max(\boldsymbol{u}_c(S^*)), \forall c \in \mathcal{C}$, *and* $S^* = \mathrm{argmax}_{S:|S| \leq k} F_\alpha(S)$ *denotes the optimal solution for the FIM problem based on $\mathcal{R}$, then $\hat{F}_\alpha(S^*, \mathcal{R}) \geq (1 - \varepsilon_1) \cdot OPT_F$ holds at least $1 - \delta_1$ probability if $\theta \geq C\theta_1$.*

**Lemma 4.** *Let* $\delta_2 \in (0, 1)$, $\varepsilon_2 = (\frac{e}{e-1})\varepsilon - \varepsilon_1$, *and* $\theta_2 = \frac{8Q^2 \ln(C\binom{n_G}{k}/\delta_2)}{\varepsilon_2^2(1-b_0)}$ *where $Q$ is the approximation parameter,* $b_0 = max(\boldsymbol{u}_c(S^\#)), \forall c \in \mathcal{C}$ *where $S^\#$ could be an arbitrary fair solution. For each bad $S$ (which indicates $F_\alpha(S) < (1 - 1/e - \varepsilon) \cdot OPT_F$, $\hat{F}_\alpha(S, \mathcal{R}) \geq (1 - 1/e)(1 - \varepsilon_1) \cdot OPT_F$ holds at most $\delta_2/\binom{n_G}{k}$ probability if $\theta \geq C\theta_2$.*

Please refer to Appendix for the detailed proof of Lemma 3 and Lemma 4.

**Theorem 1.** *For every* $\varepsilon > 0$, $\ell > 0$, $0 < \alpha < 1$, *and* $Q \geq 2$, *by setting* $\delta_1 = \delta_2 = 1/2n_G^\ell$ *and* $\theta \geq C \cdot max(\theta_1, \theta_2)$, *the output $S$ of* $\mathsf{FIMM}$ *satisfies* $F_\alpha(S) \geq (1 - 1/e - \varepsilon) F_\alpha(S^*)$, *where $S^*$ denotes the optimal solution with probability at least* $1 - 1/n_G^\ell$.

*Proof.* Combining Lemma 3 and Lemma 4, we have $\hat{F}_\alpha(S, \mathcal{R}) \geq (1 - 1/e - \varepsilon) \cdot OPT_F$ at least $1 - \delta_1 - \delta_2$ probability based on the union bound. If we set $\delta_1 = \delta_2 = 1/2n_G^\ell$, then, following the standard analysis of IMM, our $\mathsf{FIMM}$ algorithm provides $(1 - 1/e - \varepsilon)$-approximation with probability at least $1 - 1/n_G^\ell$. $\square$

If we set $\delta_1 = \delta_2 = \frac{1}{2n_G^\ell}$ and $\varepsilon_1 = \varepsilon \cdot \frac{e}{e-1} \cdot \frac{\sqrt{3}\tau_1}{\sqrt{3}\tau_1 + \sqrt{2}\tau_2}$ where $\tau_1 = \sqrt{\ln C + \ell \ln n_G + \ln 2}$ and $\tau_2^2 = \tau_1^2 + \ln\binom{n_G}{k}$, then a possible setting of $\theta$ could be $\theta = (\frac{e-1}{e})^2 \cdot \frac{4CQ^2(\sqrt{3}\tau_1 + \sqrt{2}\tau_2)^2}{\varepsilon^2(1-b_0)}$.

## 4 Experiments

### 4.1 Dataset

**Email** The Email dataset [29] is generated using email data from a large European research institution, where every node is a member of the research institution and an directed edge $(v, u)$ indicates that $v$ has sent $u$ at least one email. It contains 1,005 nodes and 25,571 directed edges. Moreover, this dataset also contains "ground-truth" community memberships of nodes, where each member belongs to exactly one of 42 departments at the research institute.

**Flixster** The Flixster dataset [30] is a network of American social movie discovery services. To transform the dataset into a weighted graph, each user is represented by a node, and a directed edge from node $u$ to $v$ is formed if $v$ rates one movie shortly after $u$ does so on the same movie. It contains 29,357 nodes and 212,614 directed edges. It also provides the learned influence probability between each node pair, which can be incorporated into the IC model. Since it has no community information, we construct the biased community structure by categorizing individuals according to their susceptibility of being influenced to highlight the level of inequality and get 100 communities. Moreover, this dataset contains the learned influence probability between each node pair, which can be incorporated into the IC model.

**Amazon** The Amazon dataset [31] is collected based on *Customers Who Bought This Item Also Bought* feature of the Amazon website. If a product $i$ is frequently co-purchased with product $j$, the graph contains an undirected edge between $i$ to $j$. The dataset also provides the ground-truth community structure which indicates the product categories. The original network has 334,863 nodes and 925,872 undirected edges. After Pruning low-quality communities (whose size is no more than 10 nodes), the Amazon network tested in our experiments 12,698 nodes, 40,096 edges, and 509 communities.

**Youtube** The Youtube dataset [31] is a network of the video-sharing web site that includes social relationships. Users form friendship each other and users can create groups which other users can join. The friendship between users is regarded as undirected edges and the user-defined groups are considered as ground-truth communities. The original network has 1,134,890 nodes and 2,987,624 undirected edges. After screening high-quality communities, it remains 30,696 nodes, 198,867 edges, and 1,157 communities.

**DBLP** The DBLP dataset [31] is the co-authorship network where two authors are connected if they ever published a paper together. Publication venues, such as journals or conferences, defines an individual ground-truth community and authors who published to a certain journal or conference form a community. The original network has 717,080 nodes and 1,049,866 undirected edges. We also perform the network pruning and finally obtain 72,875 nodes, 268,346 edges, and 1,352 communities.

### 4.2 Evaluation

Let $S_I$ denote the seed set returned by IMM [14], $S_F$ denote the seed set returned by FIMM, the performance of $S_F$ towards fairness can be evaluated via the Price of Fairness (PoF) and the Effect of Fairness (EoF) as

$$PoF = \frac{\sigma(S_I) - \sigma(S_F)}{\sigma(S_I) - k}, EoF = \left( \frac{F_\alpha(S_F) - F_\alpha(S_I)}{F_\alpha(S_I) - k} \right)^\alpha,$$

where $|S_I| = |S_F| = k$, $\sigma(\cdot)$ denotes the influence spread and $F_\alpha(\cdot)$ denotes the fair influence.

Intuitively, $PoF$ implies how much price it cost to access fairness and $EoF$ implies to what extent it steps towards fairness.

### 4.3 Results

We test IMM and our proposed FIMM algorithm in the experiment. In all tests, we run 10,000 Monte-Carlo simulations to evaluate both the influence spread and the fair influence under IC model. We also test influence probability $p$, inequality aversion parameter $\alpha$ and the seed budget $k$.

#### 4.3.1 Email & Flixster

For the Email network, we set $\alpha = 0.5$. Since the network is small, we apply the Uniformed IC model where the influence probability is the same across all edges. We test different probabilities that range

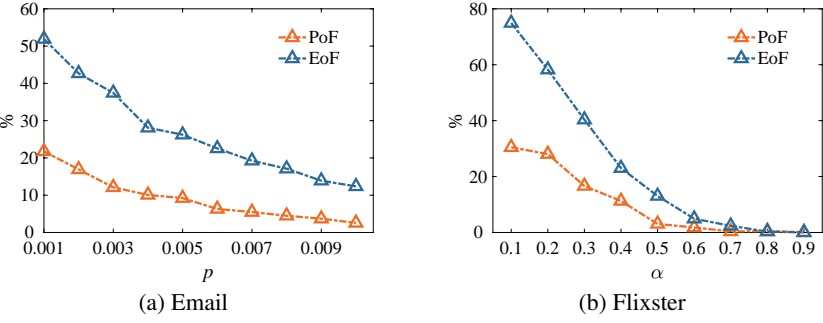

(a) Email              (b) Flixster

Figure 1: Results of testing influence probability $p$ on Email and aversion parameter $\alpha$ on Flixster.

from 0.001 to 0.01 with the step of 0.001. For the Flixster network, we test the inequality aversion parameter $\alpha$ which ranges from 0.1 to 0.9 with the step of 0.1. We set $k = 50$ for both networks and the results are shown in Figure 1.

As the influence probability $p$ increases, both $PoF$ and $EoF$ show a downward trend. This may be attributed to the increased challenges faced by disadvantaged communities in being influenced when $p$ is small. Similarly, both $PoF$ and $EoF$ also show a downward trend with the increase of the aversion parameter $\alpha$. The reason lies that communities experience greater promotions in fair influence when the aversion parameter is smaller, resulting in higher $EoF$ and $PoF$. Moreover, there is hardly any fairness when $\alpha \geq 0.7$ where the gap between $\boldsymbol{u}^\alpha$ and $\boldsymbol{u}$ is just too small.

### 4.3.2  Amazon, Youtube & DBLP

For Amazon, Youtube, and DBLP networks, we set $\alpha = 0.5$ and $p(v_i, v_j) = 1/d_{in}(v_j)$ where $d_{in}$ denotes the in-degree as the influence probability following the weighted IC model [17]. We test different seed budget $k$ that ranges from 5 to 50 with the step of 5. Results are shown in Figure 2.

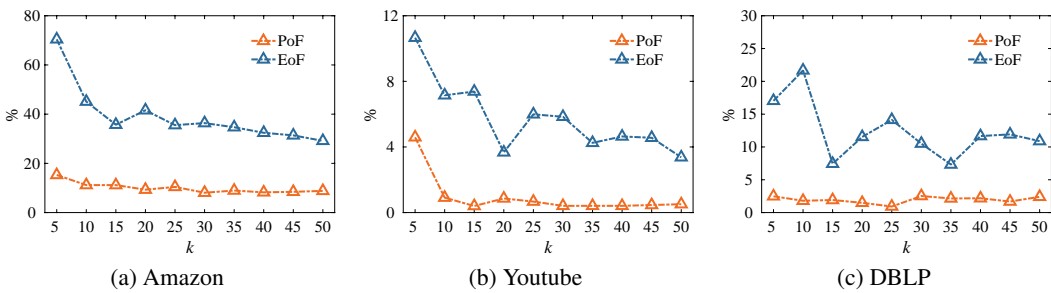

|       (a) Amazon        |       (b) Youtube       |        (c) DBLP         |

Figure 2: Results of testing seed budget $k$ on Amazon, Youtube, and DBLP.

Generally, FIMM tends to produce a noticeably fairer output when $k$ is small. It reflects the idea that enforcing fairness as a constraint becomes easier when there is an abundance of resources available. However, there are also some exceptions where smaller $k$ leads to a lower EoF, *e.g.*, $k = 20$ Figure 2(b) and $k = 5, 15$ in Figure 2(c). This may be attributed to the fact that the seed selection in FIMM follows a pattern of remedying the previously fair solutions in each round.

## 5  Conclusion

This paper focuses on the fair influence maximization problem with efficient algorithms. We first tackle the challenge of carrying out the unbiased estimation of the fractional power of the expected proportion of activated nodes in each community. Then, we deal with the challenge of integrating unbiased estimation into the RIS framework and propose an $(1 - 1/e - \varepsilon)$ approximation algorithm FIMM. We further give a theoretical analysis that addresses the concentration of the unbiased estimator of the fractional power. The experiments validate that our algorithm is both scalable and effective, which is consistent with our theoretical analysis.

There are several future directions from this research. One direction is to find some other unbiased estimators for the fair influence that would be easier to calculate through RIS. Another direction is to explore a more efficient seed selection strategy. The fairness bound is also a meaningful research direction.

**Limitation**  The limitations of our work are mainly two points. The first limitation is that our algorithm can only be efficient when the number of communities is small (e.g., a constant). The second limitation is that our algorithm is based on the assumption that the minimal community utility of an arbitrary seed set would not be too big.

## Acknowledgments and Disclosure of Funding

This work was supported by the Fundamental Research Funds for the Central Universities (No.2022QN1093).

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

## Appendix

## A Proofs

**Fact 2.** *(Chernoff bound) Let $X_1, X_2, \ldots, X_R$ be $R$ independent random variables with $X_i$ having range [0, 1], and there exists $\mu \in [0, 1]$ making $\mathbb{E}[X_i] = \mu$ for any $i \in [R]$. Let $Y = \sum_{i=1}^R X_i$, for any $\gamma > 0$,*

$$\Pr\{Y - t\mu \geq \gamma \cdot t\mu\} \leq \exp(-\frac{\gamma^2}{2 + \frac{2}{3}\gamma}t\mu).$$

*For any $0 < \gamma < 1$,*

$$\Pr\{Y - t\mu \leq -\gamma \cdot t\mu\} \leq \exp(-\frac{\gamma^2}{2}t\mu).$$

**Lemma 3.** *Let $\delta_1 \in (0, 1)$, $\varepsilon_1 \in (0, 1)$, and $\theta_1 = \frac{12Q^2 \ln(C/\delta_1)}{\varepsilon_1^2(1-b)}$ where $Q$ is the approximation parameter, $b = max(\boldsymbol{u}_c(S^*)), \forall c \in \mathcal{C}$, and $S^* = \text{argmax}_{S:|S|\leq k} F_\alpha(S)$ denotes the optimal solution for the FIM problem based on $\mathcal{R}$, then $\hat{F}_\alpha(S^*, \mathcal{R}) \geq (1 - \varepsilon_1) \cdot OPT_F$ holds at least $1 - \delta_1$ probability if $\theta \geq C\theta_1$.*

*Proof.* Let $X_c^i$ be the random variable for each $R_i \in \mathcal{R}$ ($R_i$ rooted in $c$), such that $X_c^i = 1$ if $S^* \cap R_c(i) \neq \varnothing$, and $X_c^i = 0$ otherwise. Let $\pi_c = \theta_c - \sum_{i \in [\theta_c]} X_c^i$.

$$Pr\left\{\hat{F}_\alpha(S^*, \mathcal{R}) < (1 - \varepsilon_1) \cdot OPT_F\right\}$$

$$= Pr\left\{\sum_{c \in \mathcal{C}} n_c \left(1 - \alpha \sum_{n=1}^\infty \eta(n, \alpha)\frac{(\theta_c - n)!}{\theta_c!}\left\{\sum \overline{X_c^{i_1}} \cdot \overline{X_c^{i_2}} \cdots \overline{X_c^{i_n}}\right\}\right) < (1 - \varepsilon_1) \cdot F_\alpha(S^*)\right\}$$

$$= Pr\left\{\sum_{c \in \mathcal{C}} n_c \left(1 - \alpha \sum_{n=1}^\infty \eta(n, \alpha)\frac{(\theta_c - n)!}{\theta_c!}\left\{\sum \overline{X_c^{i_1}} \cdot \overline{X_c^{i_2}} \cdots \overline{X_c^{i_n}}\right\}\right) < (1 - \varepsilon_1) \sum_{c \in \mathcal{C}} n_c \left(\boldsymbol{u}_c(S^*)\right)^\alpha\right\}$$

$$= Pr\left\{\sum_{c \in \mathcal{C}} n_c \left(1 - \alpha \sum_{n=1}^\infty \eta(n, \alpha) \prod_{i=0}^{n-1}\frac{\pi_c - i}{\theta_c - i}\right) < (1 - \varepsilon_1) \sum_{c \in \mathcal{C}} n_c \left(1 - \alpha \sum_{n=1}^\infty \eta(n, \alpha)\left(1 - \boldsymbol{u}_c(S^*)\right)^n\right)\right\}$$

$$\leq 1 - \prod_{c \in \mathcal{C}} \left(1 - Pr\left\{1 - \alpha \sum_{n=1}^\infty \eta(n, \alpha) \prod_{i=0}^{n-1}\frac{\pi_c - i}{\theta_c - i} < (1 - \varepsilon_1)\left(1 - \alpha \sum_{n=1}^\infty \eta(n, \alpha)\left(1 - \boldsymbol{u}_c(S^*)\right)^n\right)\right\}\right)$$

$$\tag{6}$$

For each community $c$, let $\varepsilon_1' = \frac{1 - \alpha \sum_{n=1}^\infty \eta(n, \alpha)\left(1 - \boldsymbol{u}_c(S^*)\right)^n}{\alpha \sum_{n=1}^\infty \eta(n, \alpha)\left(1 - \boldsymbol{u}_c(S^*)\right)^n} \cdot \varepsilon_1$, thus $\varepsilon_1' \geq \varepsilon_1$ when $\alpha \leq 1/2$, and

$$Pr\left\{1 - \alpha \sum_{n=1}^\infty \eta(n, \alpha) \prod_{i=0}^{n-1}\frac{\pi_c - i}{\theta_c - i} < (1 - \varepsilon_1)\left(1 - \alpha \sum_{n=1}^\infty \eta(n, \alpha)\left(1 - \boldsymbol{u}_c(S^*)\right)^n\right)\right\}$$

$$= Pr\left\{1 - \alpha \sum_{n=1}^\infty \eta(n, \alpha) \prod_{i=0}^{n-1}\frac{\pi_c - i}{\theta_c - i} < 1 - \alpha \sum_{n=1}^\infty \eta(n, \alpha)\left(1 - \boldsymbol{u}_c(S^*)\right)^n - \varepsilon_1\left(1 - \alpha \sum_{n=1}^\infty \eta(n, \alpha)\left(1 - \boldsymbol{u}_c(S^*)\right)^n\right)\right\}$$

$$= Pr\left\{\alpha \sum_{n=1}^\infty \eta(n, \alpha) \prod_{i=0}^{n-1}\frac{\pi_c - i}{\theta_c - i} > \alpha \sum_{n=1}^\infty \eta(n, \alpha)\left(1 - \boldsymbol{u}_c(S^*)\right)^n + \varepsilon_1\left(1 - \alpha \sum_{n=1}^\infty \eta(n, \alpha)\left(1 - \boldsymbol{u}_c(S^*)\right)^n\right)\right\}$$

$$= Pr\left\{\sum_{n=1}^\infty \eta(n, \alpha) \prod_{i=0}^{n-1}\frac{\pi_c - i}{\theta_c - i} > (1 + \varepsilon_1') \sum_{n=1}^\infty \eta(n, \alpha)\left(1 - \boldsymbol{u}_c(S^*)\right)^n\right\}$$

$$\leq Pr\left\{\sum_{n=1}^{\pi_c} \eta(n, \alpha) \prod_{i=0}^{n-1}\frac{\pi_c - i}{\theta_c - i} > (1 + \varepsilon_1') \sum_{n=1}^{\pi_c} \eta(n, \alpha)\left(1 - \boldsymbol{u}_c(S^*)\right)^n\right\} \tag{7}$$

$$\leq Pr\left\{\sum_{n=1}^{\pi_c}\eta(n,\alpha)(\frac{\pi_c}{\theta})^n > (1+\varepsilon_1')\sum_{n=1}^{\pi_c}\eta(n,\alpha)\big(1-\boldsymbol{u}_c(S^*)\big)^n\right\}$$

$$\leq 1 - Pr\left\{(\frac{\pi_c}{\theta})^{\pi_c} < (1+\varepsilon_1')\big(1-\boldsymbol{u}_c(S^*)\big)^{\pi_c}\right\}$$

$$= Pr\left\{(\frac{\pi_c}{\theta})^{\pi_c} \geq (1+\varepsilon_1')\big(1-\boldsymbol{u}_c(S^*)\big)^{\pi_c}\right\}$$

$$\left(\text{Let } 1+\varepsilon_0 = \sqrt[\pi_c]{1+\varepsilon_1'}\right)$$

$$= Pr\left\{\frac{\pi_c}{\theta} \geq (1+\varepsilon_0)\big(1-\boldsymbol{u}_c(S^*)\big)\right\}$$

$$= \Pr\left\{\pi_c - \theta_c\big(1-\boldsymbol{u}_c(S^*)\big) \geq \varepsilon_0\theta_c\big(1-\boldsymbol{u}_c(S^*)\big)\right\}$$

$$\leq exp\left(-\frac{\varepsilon_0^2}{3}\theta_c\big(1-\boldsymbol{u}_c(S^*)\big)\right) \tag{8}$$

Since $0 \leq \frac{\epsilon}{2x} \leq \sqrt[x]{1+\epsilon} - 1 \leq \frac{\epsilon}{x}$ for $0 \leq \epsilon \leq 1$ and $x \geq 1$, it holds $\frac{2x}{\epsilon} \geq \frac{1}{\sqrt[x]{1+\epsilon}-1}$. Let $\theta_c \geq \frac{12\pi_c^2\ln(C/\delta_1)}{\varepsilon_1^2\big(1-\boldsymbol{u}_c(S^*)\big)} \geq \frac{3\ln(C/\delta_1)}{\big(\sqrt[\pi_c]{1+\varepsilon_1'}-1\big)^2\big(1-\boldsymbol{u}_c(S^*)\big)} = \frac{3\ln(C/\delta_1)}{\varepsilon_0^2\big(1-\boldsymbol{u}_c(S^*)\big)}$, then

$$Eq.\ 8 = exp\left(-\frac{\varepsilon_0^2}{3}\theta_c\big(1-\boldsymbol{u}_c(S^*)\big)\right)$$

$$\leq exp\left(-\frac{\varepsilon_0^2}{3}\frac{3\ln(C/\delta_1)}{\varepsilon_0^2\big(1-\boldsymbol{u}_c(S^*)\big)}\big(1-\boldsymbol{u}_c(S^*)\big)\right)$$

$$= \delta_1/C \tag{9}$$

Therefore,

$$Eq.(6) \leq 1 - \prod_{c\in\mathcal{C}}(1-\delta_1/C) \leq \delta_1 \tag{10}$$

To limit Eq.(7) to the first $Q$ ($Q \geq 2$) terms, Eq.(7) becomes

$$Pr\left\{\sum_{n=1}^{Q}\eta(n,\alpha)\prod_{i=0}^{n-1}\frac{\pi_c-i}{\theta_c-i} > (1+\varepsilon_1')\sum_{n=1}^{\pi_c}\eta(n,\alpha)\big(1-\boldsymbol{u}_c(S^*)\big)^n\right\}$$

$$\leq Pr\left\{\sum_{n=1}^{Q}\eta(n,\alpha)(\frac{\pi_c}{\theta})^n > (1+\varepsilon_1')\sum_{n=1}^{\pi_c}\eta(n,\alpha)\big(1-\boldsymbol{u}_c(S^*)\big)^n\right\}$$

$$\leq 1 - Pr\left\{(\frac{\pi_c}{\theta})^Q < (1+\varepsilon_1')\big(1-\boldsymbol{u}_c(S^*)\big)^Q\right\}$$

$$\leq Pr\left\{(\frac{\pi_c}{\theta})^Q \geq (1+\varepsilon_1')\big(1-\boldsymbol{u}_c(S^*)\big)^Q\right\}$$

$$\left(\text{Let } 1+\varepsilon_0 = \sqrt[Q]{1+\varepsilon_1'}\right)$$

$$= \Pr\left\{\pi_c - \theta_c\big(1-\boldsymbol{u}_c(S^*)\big) \geq \varepsilon_0\theta_c\big(1-\boldsymbol{u}_c(S^*)\big)\right\}$$

$$\leq exp\left(-\frac{\varepsilon_0^2}{3}\theta_c\big(1-\boldsymbol{u}_c(S^*)\big)\right) \tag{11}$$

$$\left(\text{Let } \theta_c \geq \frac{12Q^2\ln(C/\delta_1)}{\varepsilon_1^2\big(1-\boldsymbol{u}_c(S^*)\big)} \geq \frac{3\ln(C/\delta_1)}{\big(\sqrt[Q]{1+\varepsilon_1'}-1\big)^2\big(1-\boldsymbol{u}_c(S^*)\big)} = \frac{3\ln(C/\delta_1)}{\varepsilon_0^2\big(1-\boldsymbol{u}_c(S^*)\big)}\right)$$

$$\leq \delta_1/C \tag{12}$$

Therefore,

$$Eq.(6) \leq \delta_1 \tag{13}$$

It indicates $Pr\left\{\hat{F}_\alpha(S^*,\mathcal{R}) < (1-\varepsilon_1)\cdot OPT_F\right\} \geq 1 - \delta_1$, thus concludes the proof. $\qquad\square$

**Lemma 4.** *Let $\delta_2 \in (0,1)$, $\varepsilon_2 = (\frac{e}{e-1})\varepsilon - \varepsilon_1$, and $\theta_2 = \frac{8Q^2 \ln(C\binom{n_G}{k}/\delta_2)}{\varepsilon_2^2(1-b_0)}$ where $Q$ is the approxima-
tion parameter, $b_0 = max(\boldsymbol{u}_c(S^\#)), \forall c \in \mathcal{C}$ where $S^\#$ could be an arbitrary fair solution. For each
bad $S$ (which indicates $F_\alpha(S) < (1 - 1/e - \varepsilon) \cdot OPT_F$, $\hat{F}_\alpha(S, \mathcal{R}) \geq (1 - 1/e)(1 - \varepsilon_1) \cdot OPT_F$
holds at most $\delta_2/\binom{n_G}{k}$ probability if $\theta \geq C\theta_2$.*

*Proof.* Let $X_c^i$ be the random variable for each $R_i \in \mathcal{R}$ ($R_i$ rooted in $c$), such that $X_c^i = 1$ if
$S \cap R_c(i) \neq \varnothing$, and $X_c^i = 0$ otherwise. Let $\pi_c = \theta_c - \sum_{i \in [\theta_c]} X_c^i$.

$$Pr\left\{\hat{F}_\alpha(S, \mathcal{R}) \geq (1 - \frac{1}{e})(1 - \varepsilon_1) \cdot OPT_F\right\}$$

$$\leq Pr\left\{\hat{F}_\alpha(S, \mathcal{R}) \geq (1 + \varepsilon_2)F_\alpha(S)\right\}$$

$$= Pr\left\{\sum_{c \in \mathcal{C}} n_c \left(1 - \alpha \sum_{n=1}^{\infty} \eta(n, \alpha) \prod_{i=0}^{n-1} \frac{\pi_c - i}{\theta_c - i}\right) \geq (1 + \varepsilon_2) \sum_{c \in \mathcal{C}} n_c \left(1 - \alpha \sum_{n=1}^{\infty} \eta(n, \alpha)\left(1 - \boldsymbol{u}_c(S)\right)^n\right)\right\}$$

$$\leq 1 - \prod_{c \in \mathcal{C}} \left(1 - Pr\left\{1 - \alpha \sum_{n=1}^{\infty} \eta(n, \alpha) \prod_{i=0}^{n-1} \frac{\pi_c - i}{\theta_c - i} \geq (1 + \varepsilon_2)\left(1 - \alpha \sum_{n=1}^{\infty} \eta(n, \alpha)\left(1 - \boldsymbol{u}_c(S)\right)^n\right)\right\}\right)$$

$$\tag{14}$$

For each community $c$, let $\varepsilon_2' = \frac{1 - \alpha \sum_{n=1}^{\infty} \eta(n,\alpha)\left(1 - \boldsymbol{u}_c(S)\right)^n}{\alpha \sum_{n=1}^{\infty} \eta(n,\alpha)\left(1 - \boldsymbol{u}_c(S)\right)^n} \cdot \varepsilon_2$, thus $\varepsilon_2' \geq \varepsilon_2$ when $\alpha \leq 1/2$, and

$$Pr\left\{1 - \alpha \sum_{n=1}^{\infty} \eta(n, \alpha) \prod_{i=0}^{n-1} \frac{\pi_c - i}{\theta_c - i} \geq (1 + \varepsilon_2)\left(1 - \alpha \sum_{n=1}^{\infty} \eta(n, \alpha)\left(1 - \boldsymbol{u}_c(S)\right)^n\right)\right\}$$

$$= Pr\left\{1 - \alpha \sum_{n=1}^{\infty} \eta(n, \alpha) \prod_{i=0}^{n-1} \frac{\pi_c - i}{\theta_c - i} \geq 1 - \alpha \sum_{n=1}^{\infty} \eta(n, \alpha)\left(1 - \boldsymbol{u}_c(S)\right)^n + \varepsilon_2\left(1 - \alpha \sum_{n=1}^{\infty} \eta(n, \alpha)\left(1 - \boldsymbol{u}_c(S)\right)^n\right)\right\}$$

$$= Pr\left\{\alpha \sum_{n=1}^{\infty} \eta(n, \alpha) \prod_{i=0}^{n-1} \frac{\pi_c - i}{\theta_c - i} \leq \alpha \sum_{n=1}^{\infty} \eta(n, \alpha)\left(1 - \boldsymbol{u}_c(S)\right)^n - \varepsilon_2\left(1 - \alpha \sum_{n=1}^{\infty} \eta(n, \alpha)\left(1 - \boldsymbol{u}_c(S)\right)^n\right)\right\}$$

$$= Pr\left\{\sum_{n=1}^{\infty} \eta(n, \alpha) \prod_{i=0}^{n-1} \frac{\pi_c - i}{\theta_c - i} \leq (1 - \varepsilon_2') \sum_{n=1}^{\infty} \eta(n, \alpha)\left(1 - \boldsymbol{u}_c(S)\right)^n\right\} \tag{15}$$

To limit Eq. 15 to the first $Q$ ($Q \geq 2$) terms, let $y = \frac{(1 - \boldsymbol{u}_c(S))^{Q+1}}{(Q+1)\boldsymbol{u}_c(S)}$, $x = \frac{y\theta_c^2}{\theta_c - \pi_c + y\theta_c}$, Eq. 15 becomes

$$Pr\left\{\sum_{n=1}^{Q} \eta(n, \alpha) \prod_{i=0}^{n-1} \frac{\pi_c - i}{\theta_c - i} \leq (1 - \varepsilon_2') \sum_{n=1}^{\infty} \eta(n, \alpha)\left(1 - \boldsymbol{u}_c(S)\right)^n\right\}$$

$$= Pr\left\{\frac{\pi_c}{\theta_c} - (1 - \varepsilon_2') \sum_{n=Q+1}^{\infty} \eta(n, \alpha)\left(1 - \boldsymbol{u}_c(S)\right)^n + \sum_{n=2}^{Q} \eta(n, \alpha) \prod_{i=0}^{n-1} \frac{\pi_c - i}{\theta_c - i} \leq (1 - \varepsilon_2') \sum_{n=1}^{Q} \eta(n, \alpha)\left(1 - \boldsymbol{u}_c(S)\right)^n\right\}$$

$$\leq Pr\left\{\frac{\pi_c}{\theta_c} - y + \sum_{n=2}^{Q} \eta(n, \alpha) \prod_{i=0}^{n-1} \frac{\pi_c - i}{\theta_c - i} \leq (1 - \varepsilon_2') \sum_{n=1}^{Q} \eta(n, \alpha)\left(1 - \boldsymbol{u}_c(S)\right)^n\right\}$$

$$\leq Pr\left\{\frac{\pi_c - x}{\theta_c - x} + \sum_{n=2}^{Q} \eta(n, \alpha) \prod_{i=0}^{n-1} \frac{\pi_c - i}{\theta_c - i} \leq (1 - \varepsilon_2') \sum_{n=1}^{Q} \eta(n, \alpha)\left(1 - \boldsymbol{u}_c(S)\right)^n\right\}$$

$$\leq Pr\left\{\sum_{n=1}^{Q} \eta(n, \alpha)(\frac{\pi_c - Q + 1}{\theta_c - Q + 1})^n \leq (1 - \varepsilon_2') \sum_{n=1}^{Q} \eta(n, \alpha)\left(1 - \boldsymbol{u}_c(S)\right)^n\right\} \text{ (when } x \leq Q - 1)$$

$$\leq 1 - Pr\left\{(\frac{\pi_c - Q + 1}{\theta_c - Q + 1})^Q > (1 - \varepsilon_2')\left(1 - \boldsymbol{u}_c(S)\right)^Q\right\}$$

$$= Pr\left\{(\frac{\pi_c - Q + 1}{\theta_c - Q + 1})^Q \leq (1 - \varepsilon_2')(1 - \boldsymbol{u}_c(S))^Q\right\}$$

$$\left(\text{Let } 1 - \varepsilon_0 = \sqrt[Q]{1 - \varepsilon_2'}\right)$$

$$= Pr\left\{\frac{\pi_c - Q + 1}{\theta_c - Q + 1} \leq (1 - \varepsilon_0)(1 - \boldsymbol{u}_c(S))\right\} \quad (16)$$

Let $\varepsilon_0' = \varepsilon_0 + \frac{\theta_c}{\pi_c}\frac{\pi_c - Q + 1}{\theta_c - Q + 1} - 1, \varepsilon_0 \geq \varepsilon_0' \geq 1 - \frac{\theta_c}{\pi_c}\frac{\pi_c - Q}{\theta_c - Q}$, Eq. 16 becomes

$$Pr\left\{\frac{\pi_c - Q + 1}{\theta_c - Q + 1} \leq (1 - \varepsilon_0)(1 - \boldsymbol{u}_c(S))\right\}$$

$$= Pr\left\{\frac{\pi_c}{\theta_c} \leq (1 - \frac{\pi_c}{\theta_c}\frac{\theta_c - Q + 1}{\pi_c - Q + 1}\varepsilon_0')(1 - \boldsymbol{u}_c(S))\right\}$$

$$\leq exp\left(-\frac{(\frac{\pi_c}{\theta_c}\frac{\theta_c - Q + 1}{\pi_c - Q + 1}\varepsilon_0')^2}{2}\theta_c(1 - \boldsymbol{u}_c(S))\right)$$

$$\leq exp\left(-\frac{\varepsilon_0'^2}{2}\theta_c(1 - \boldsymbol{u}_c(S))\right)$$

$$\left(\text{Let } \theta_c \geq \frac{8Q^2\ln(C\binom{n_G}{k}/\delta_2)}{\varepsilon_2^2(1 - \boldsymbol{u}_c(S))} \geq \frac{8\ln(C\binom{n_G}{k}/\delta_2)}{\left(1 - \sqrt[Q]{1 - \varepsilon_2'}\right)^2(1 - \boldsymbol{u}_c(S))} \geq \frac{2\ln(C\binom{n_G}{k}/\delta_2)}{\varepsilon_0'^2(1 - \boldsymbol{u}_c(S))}\right)$$

$$\leq \delta_2/C\binom{n_G}{k} \quad (17)$$

Therefore,

$$Eq.(14) \leq 1 - \prod_{c\in\mathcal{C}}(1 - \delta_2/C\binom{n_G}{k}) \leq \delta_2/\binom{n_G}{k} \quad (18)$$

It indicates $Pr\left\{\hat{F}_\alpha(S, \mathcal{R}) \geq (1 - 1/e)(1 - \varepsilon_1) \cdot OPT_F\right\} \leq \delta_2/\binom{n_G}{k}$, thus concludes the proof.

$\square$

## B  Additional Experiments

### B.1  Running Time

The number of RR sets is mainly determined by both the size of the network and the number of communities. In the following, we exhibit the runtime of our algorithm with the scale of networks. Note that our algorithm is currently implemented in Matlab 2022a, thus it costs more time to generate RR sets (generating RR sets in C++ could be at least 100 times faster). **RRsets** refers to the time (seconds) used to generate RR sets, **IMM** and **FIMM** denote the time used to select seeds based on the generated RR sets for IMM and FIMM, respectively.

Table 2: Running time (seconds).

| Network | $n_G$ | $C$ | RRsets | IMM | FIMM |
|---------|-------|-----|--------|-----|------|
| Email | 1,005 | 42 | 14.281 | 0.011 | 0.020 |
| Amazon | 12,698 | 509 | 82.204 | 0.028 | 0.283 |
| Youtube | 30,696 | 1,157 | 162.533 | 0.052 | 3.592 |
| DBLP | 72,875 | 1,352 | 735.755 | 0.063 | 18.259 |

Note that the proposed algorithm is more suitable and could be much more efficient under scenarios where the number of communities is rather limited (*e.g.*, a constant).

## B.2 Comparison with Equality-based methods

Our proposed algorithm is based on the notion of welfare fairness, which is in favor of fair result-aware seeding. In this subsection, we aim to explore the difference between such results and community-aware seeding, which is based on the notion of equality. The Equality asks to divide the budget $k$ proportionally to the cluster sizes, *i.e.*, $|S \cap V_c| \approx k \cdot n_c / n_G$. We adopt two strategies to select seeds under Equality: picking nodes with the highest degree per community and running IMM inside each community.

The dataset tested is the Email network (since the number of communities of other datasets is more than 50 and $k$ is set to 50). The experiment settings are the same as the setting for Email in our Section 4 ($\alpha = 0.5$ and $p$ ranges from 0.001 to 0.01 with the step of 0.001). The Table 3 below exhibits the results, where $S_{ours}$, $S_{\text{C-HD}}$ and $S_{\text{C-IMM}}$ refer to seeds selected by our method, seeds selected by community-aware highest degree, and community-aware IMM, respectively. $PoF$ and $EoF$ are calculated w.r.t IMM in the classic influence maximization problem.

Table 3: Comparison with equality-based methods

| $p$ | $PoF$ | | | $EoF$ | | |
|---|---|---|---|---|---|---|
| | $S_{ours}$ | $S_{\text{C-HD}}$ | $S_{\text{C-IMM}}$ | $S_{ours}$ | $S_{\text{C-HD}}$ | $S_{\text{C-IMM}}$ |
| 0.001 | 21.77% | 31.99% | 21.56% | 51.91% | 45.08% | 45.77% |
| 0.002 | 16.92% | 31.50% | 20.56% | 42.68% | 37.31% | 38.91% |
| 0.003 | 12.11% | 30.73% | 20.76% | 37.44% | 31.76% | 34.23% |
| 0.004 | 10.08% | 29.63% | 19.11% | 28.10% | 20.97% | 25.94% |
| 0.005 | 9.22% | 28.85% | 17.63% | 26.23% | 16.49% | 23.63% |
| 0.006 | 6.31% | 28.25% | 17.96% | 22.54% | 3.82% | 18.06% |
| 0.007 | 5.48% | 27.06% | 16.21% | 19.25% | N/A | 13.57% |
| 0.008 | 4.49% | 26.03% | 15.77% | 17.11% | N/A | 7.99% |
| 0.009 | 3.70% | 24.65% | 14.88% | 13.89% | N/A | N/A |
| 0.01 | 2.57% | 24.07% | 15.64% | 12.37% | N/A | N/A |

As can be seen from the results, both $PoF$ (the lower the better) and $RoF$ of $S_{ours}$ (the higher the better) are always better than that of both $S_{\text{C-HD}}$ and $S_{\text{C-IMM}}$. In other words, $S_{\text{C-HD}}$ pays more price of fairness yet achieves a lower degree of fairness. Compared with $S_{\text{C-HD}}$, $S_{\text{C-IMM}}$ yields a better performance, Moreover, as $p$ increases, the fair influence of both $S_{\text{C-HD}}$ and $S_{\text{C-IMM}}$ is even lower (leading to N/A of $EoF$) than that of IMM, which does not even contribute to fairness at all. The reason is that community-aware seeding highlights fairness in the process of seed allocation but not selection, while welfare fairness (also, MMF and DC) highlights fairness in the spreading results. The former could have an explicit fair distribution in seeding, but may still lead to unfair results.

