# OpenReview forum: "Scalable Fair Influence Maximization"
_NeurIPS.cc/2023/Conference — NeurIPS 2023 poster_

### Official Review · Reviewer_uZjd · 2023-07-01

**Soundness:** 4 excellent
**Presentation:** 3 good
**Contribution:** 3 good
**Rating:** 6
**Confidence:** 5

**Summary:**

The paper proposes an algorithm for fair influence maximization. The objective is defined as a sum of powers of a utility function for each cluster, indicating that each cluster should receive some influence. Reducing the exponent will bring the objective to a fairer distribution of influence. An algorithm based on max cover on reverse reachable sets is presented. Optimal number of sets are derived to guarantee $(1-1/e - \epsilon)$-approximation

**Strengths:**

- The key idea to convert the powers into sums using the binomial theorem is elegant, as it helps with modifying the existing algorithm on influence maximization for the proposed objective.
- Detailed derivations are presented.

**Weaknesses:**

- Baselines: While the theoretical guarantee is the key contribution, the experiment part of the paper can be strengthened. How much do we gain from this algorithm compared to some simple heuristics? One may consider simple heuristics like degree. Better heuristics may also be generated, e.g., (1) picking high-degree nodes from each cluster, and (2) using the RR sets in Algorithm 1 but performing max-cover without Eq (5) [maybe similar to performing Influence Maximization independently in each cluster]

- The derivation assumes that a fair solution exists, in the form of $b = \max \mathbf{u_c} (S^{\\#})$. The usage of this appears in the proofs, but some indication of how this would impact the main results could be discussed.

- The symbol $n$ is overloaded. It is used as an index for the summation as well as the number of nodes. I think the paper will benefit from a table of symbols.


**Questions:**

- Please see the issues raised in "Weaknesses"
- Do we need the constraint $\alpha > 0$? My intuition is that the method will work with negative $\alpha$ as well, but it needs to be carefully analyzed.

**Limitations:**

Minor issue: A discussion on the impact of the assumptions could be included.

---

> ### Author Rebuttal · Authors · 2023-08-05
>
> ### **Response to Reviewer uZjd**
>
> Thank you for your insightful comments and suggestions. We reply to all your comments below.
>
> > **Weakness 1**: Baselines: While the theoretical guarantee is the key contribution, the experiment part of the paper can be strengthened. How much do we gain from this algorithm compared to some simple heuristics? One may consider simple heuristics like degree. Better heuristics may also be generated, e.g., (1) picking high-degree nodes from each cluster, and (2) using the RR sets in Algorithm 1 but performing max-cover without Eq (5) [maybe similar to performing Influence Maximization independently in each cluster]
>
> **A1**: Thanks for addressing an interesting problem. However, this paper focuses on designing an efficient algorithm for the fair influence maximization problem based on the notion of welfare fairness. Therefore, adopting simple heuristics as degree can hardly achieve the fairness that the problem asks for. Besides, we have adopted IMM (just as using the RR sets in Algorithm 1 but performing max-cover without Eq (5)) as our baseline, which is how we calculate RoF and PoF. In other words, we test how much influence spread would cost to access fairness and to what extent our algorithm steps towards fairness, compared with IMM.
>
> >**Weakness 2**: The derivation assumes that a fair solution exists, in the form of $b_0=\max \bf{u}_c(S^\\#)$. The usage of this appears in the proofs, but some indication of how this would impact the main results could be discussed.\
> >**Limitations**: Minor issue: A discussion on the impact of the assumptions could be included.
>
> **A2**: We were sorry for our previous sloppy expression that claims $S^\\#$ as a fair solution. Actually, we intend to assume that the $b_0=\max \bf{u}_c(S^\\#)$ would not be too large for an arbitrary solution $S^\\#$ under the fair influence maximization problem. The $b$ would impact the number of RR sets we need to generate to guarantee our algorithm. As shown in Lemma 4, the bigger the $b_0$, the more RR sets we need to generate. Therefore, we assume that $b_0$ should not be too large (e.g. $b_0\leq 0.9$) for an arbitrary solution. We believe that our assumption is valid in the vast majority of cases when the seed budget $k$ is small (usually 50) and the network is large with a limited number of communities.
>
>
> >**Weakness 3**: The symbol $n$ is overloaded. It is used as an index for the summation as well as the number of nodes. I think the paper will benefit from a table of symbols.
>
> **A3**: Thanks for pointing out this issue as well as your kind suggestion. We maintain the index for summation as $n$ and change the number of nodes in $G$ to $n_G$.
>
> Besides, we have summarized important symbols in our paper and presented them in the following Table:
>
> | Symbol | Explanation |
> |--|--|
> | $G=(V, E, p)$ | network |
> | $V$ | node set of the network |
> | $E$ | edge set of the network |
> | $n_G$ | number of nodes in $G$, i.e. $n_G=\|V\|$ |
> | $p(u,v)$ | probability that $u$ influence $v$ |
> | $S$ | seed set |
> | $S^*$ | the optimal seed set for fair influence maximization |
> | $ap(v,S)$ | the expected probability that $v$ is activated by $S$ |
> | $\sigma(S)$  | influence spread of $S$, i.e. $\sigma(S) = \sum_{v\in V} ap(v,S)$ |
> | $\mathcal{C}=\\{c_1,c_2,c_3,\cdots\\}$  | community structure |
> | $V_c$ | node set in community $c$ |
> | $n_c$ | number of nodes in community $c$, i.e. $n_c = \|V_c\|$ |
> | $\bf{u}_c$ | utility of $c$ (expected fraction of influenced nodes in $c$) |
> | $F_\alpha(S)$ | fair influence of $S$ |
> | $\mathcal{R}$ | the set of RR sets |
> | $\mathcal{R}_c$ | the set of RR sets rooted in community $c$ |
> | $\hat{F}_\alpha(S,\mathcal{R})$ | the unbiased estimator for fair influence of $S$ based on $\mathcal{R}$|
> | $\theta$ | the number of RR sets |
> | $\theta_c$ | the number of RR sets rooted in community $c$ |
> | $\alpha$ | aversion parameter regarding fairness |
> | $Q$ | approximation parameter for Taylor expansion |
> | $\varepsilon$ | accuracy parameter |
> | $\ell$ | confidence parameter |
>
>
> >**Question 1**: Do we need the constraint $\alpha>0$? My intuition is that the method will work with negative $\alpha$ as well, but it needs to be carefully analyzed.
>
> **A4**: Yes, we have to constraint $\alpha>0$ in our paper. Though our method may work with a negative $\alpha$, yet a negative $\alpha$ will make the fair utility $\textbf{u}_x(S)^{\alpha}>\textbf{u}_y(S)^{\alpha}$ when $\textbf{u}_x(S) < \textbf{u}_y(S)$ where $x$ and $y$ are two communities. It means that the community with a lower utility will have a higher fair utility, i.e., monotone decreasing rather than monotone increasing. Then, the algorithm would encourage communities to pursue lower utility, which is contrary to the aim of influence maximization.

---

> > ### Comment · Reviewer_uZjd · 2023-08-17
> >
> > Thanks for responding to the issues raised by me. My main concern regarding the baseline still stands. Yes, the Price and Effect of Fairness are useful measures to compare against IMM. My suggestion was to use a heuristic that is aware of the clusters. For instance, divide the budget $k$ proportionally to the cluster sizes. Then pick the highest degree nodes per cluster or by IMM per cluster.

---

> > > ### Author Response · Authors · 2023-08-20
> > > **Further Response to Reviewer uZjd**
> > >
> > > Thanks for your comments and constructive suggestion. We are sorry for our late response since we have been conducting additional experiments those days.
> > >
> > > As you suggested, we divide the budget $k$ proportionally to the cluster sizes and pick the highest degree nodes per cluster. Since the datasets used in our paper have at least 42 communities (Email network) and $k$ is set to 50, performing cluster-inside IMM is nearly the same as selecting nodes with the highest degree in each cluster. Therefore, we only show the comparison results between the seeds selected by our method and seeds selected based on the cluster-aware highest degree (just as you suggested).
> > >
> > > The dataset tested is the Email network (since the number of communities of other datasets is more than 50). The experiment settings are the same as the setting for Email in our paper ($\alpha=0.5$ and $p$ ranges from 0.001 to 0.01 with the step of 0.001). Below are the results, where $S_{ours}$ and $S_{De}$ refer to seeds selected by our method and seeds selected by cluster-aware highest degree, respectively. $PoF$ and $EoF$ are calculated w.r.t IMM in the classic Influence Maximization problem.
> > >
> > > | $p$ | $\sigma(S_{ours})$ | $\sigma(S_{De})$ | $F_\alpha(S_{ours})$ | $F_\alpha(S_{De})$ | $PoF(S_{ours})$ | $PoF(S_{De})$ | $EoF(S_{ours})$ | $EoF(S_{De})$ |
> > > |-:|-:|-:|-:|-:|-:|-:|-:|-:|
> > > 0.001|54.22|53.59 |228.94|218.08 |21.77\%|33.53\%|51.91\%|40.18\%
> > > 0.002|59.23|57.45 |239.26|227.15 |16.92\%|32.96\%|42.68\%|31.03\%
> > > 0.003|65.01|61.58 |249.95|236.36 |12.11\%|32.18\%|37.44\%|24.29\%
> > > 0.004|71.07|66.11 |260.91|246.01 |10.08\%|31.26\%|28.10\%|5.23\%
> > > 0.005|77.19|71.01 |272.28|255.97 |9.22\%|29.87\%|26.23\%|N/A
> > > 0.006|84.79|76.19 |284.68|266.02 |6.31\%|29.47\%|22.54\%|N/A
> > > 0.007|92.01|81.83 |296.82|276.53 |5.48\%|28.36\%|19.25\%|N/A
> > > 0.008|99.58|87.88 |309.17|287.35 |4.49\%|27.03\%|17.11\%|N/A
> > > 0.009|107.53|94.38 |321.94|298.47 |3.70\%|25.71\%|13.89\%|N/A
> > > 0.01|116.67|101.31 |334.08|309.91 |2.57\%|25.01\%|12.37\%|N/A
> > >
> > > As can be seen from the results, both influence spread and fair influence of $S_{ours}$ is always higher than that of $S_{De}$. In other words, $S_{De}$ pays more price of fairness yet achieves a less degree of fairness compared with $S_{ours}$. Specifically, as $p$ increases, the fair influence of $S_{De}$ is even lower (negative $EoF$) than that of IMM, which does not even consider fairness at all. The reason is that **community-aware seeding highlights fairness in selecting seeds**, while welfare fairness (also, MMF and DC) highlights fairness in the spreading results. The former could have an explicit fair distribution in seeding, but may still lead to unfair results.
> > >
> > > At last, thanks for your constructive suggestion, we believe it would be both interesting and meaningful to discuss such differences between community-aware seeding and fair result-aware seeding. If possible, we would like to report such results (as well as other datasets) in our revised paper.

---

> > > > ### Comment · Reviewer_uZjd · 2023-08-20
> > > >
> > > > Thanks for implementing this and showing the comparison.
> > > > > ... performing cluster-inside IMM is nearly the same as selecting nodes with the highest degree in each cluster
> > > > - Does this mean you selected 1 node per cluster? This would not be a fair comparison because then we are comparing 42 seeds against 50 seeds. Perhaps I misunderstood.
> > > > - It may be true that for one or two nodes IMM is similar to the highest degree, but given that the Email network is extremely small, I would encourage to actually run IMM. There are two possible variations of the heuristic. Both select nodes in a cluster-budget aware manner (1 or 2 per cluster for Email network). But for coverage from RR sets, one could look at (i) within cluster coverage only for selecting nodes or (ii) coverage in the entire network.
> > > >
> > > > I apologize for continuing this thread. But addressing this will definitely improve my rating.
> > > >
> > > > Minor issue: Why is $EoF$ define with an exponent $\alpha$? How do you get a real value if the base is negative?

---

> > > > > ### Author Response · Authors · 2023-08-21
> > > > > **Thanks for your response**
> > > > >
> > > > > Thanks again for your insightful comments and constructive suggestions. We reply to your concerns below.
> > > > >
> > > > > > **Q1**: Does this mean you selected 1 node per cluster? This would not be a fair comparison because then we are comparing 42 seeds against 50 seeds. Perhaps I misunderstood.
> > > > >
> > > > > **A1**: We are sorry that we did not state this clearly. We intended to express that when the seed budget (proportional to the cluster sizes) for a community is small, The seeds selected by IMM are pretty close to selecting the nodes with the highest degree.
> > > > >
> > > > > > **Q2**: It may be true that for one or two nodes IMM is similar to the highest degree, but given that the Email network is extremely small, I would encourage to actually run IMM. There are two possible variations of the heuristic. Both select nodes in a cluster-budget aware manner (1 or 2 per cluster for Email network). But for coverage from RR sets, one could look at (i) within cluster coverage only for selecting nodes or (ii) coverage in the entire network.
> > > > >
> > > > > **A2**: Thanks for your advice. We are now implementing the cluster-budget aware IMM with two variations as you suggested: (i) inner-cluster coverage, and (ii) whole-network coverage.
> > > > >
> > > > > However, due to the time limit, we may not be able to present related results now, But **we will definitely conduct these experiments and report such results in our revised paper**.
> > > > >
> > > > >
> > > > > > **Q3**: Minor issue: Why is $EoF$ define with an exponent $\alpha$? How do you get a real value if the base is negative?
> > > > >
> > > > > **A3**: The $EoF$ is calculated based on the fair influence, which has the parameter $\alpha$. To be exact, $F_\alpha = \sum_{c \in C} n_c u_c^\alpha$. Therefore, having an exponent $\alpha$ in $EoF$ could lead to a more obvious difference between different methods, especially when $\alpha$ is small and all methods could have a huge fair influence.
> > > > >
> > > > > Moreover, we are sorry for our carelessness in forgetting to calculate $EoF$ with the exponent in the last response. The $EoF$ of $S_{De}$ should be 40.18%, 31.03%, 24.29%, 5.23%, N/A, N/A, N/A, N/A, N/A, and N/A, where the N/A means it is less fair compared with standard IMM (as you mentioned, negative).  We have updated our previous rebuttal with the corrected $EoF$. Thank you for pointing out this problem!

---

### Official Review · Reviewer_RYE8 · 2023-07-07

**Soundness:** 3 good
**Presentation:** 3 good
**Contribution:** 3 good
**Rating:** 6
**Confidence:** 3

**Summary:**

The authors apply fairness to the influence maximization problem (IM). On a social network, IM models which node subset should be used to trigger the spread of information to maximize its effects. A motivating application is the selection of leaders for natural disaster preparedness, where in existing methods, minority leaders are disproportionately underrepresented. They are not the first to do this, however they are the first to introduce efficient algorithms for solving IM with welfare fairness, which allows the user to parametrize an objection on sliding scale between total fairness and no fairness at all (i.e., maximum influence spread).

They work in the Independent Cascade model of influence spreading, where each node has a given probability of influencing an adjacent node, and influence is randomly spread according to these probabilities across a number of timesteps. The influence spread, i.e. the objective, is the number of influenced nodes when the process converges. By submodularity, the unfair problem can be solved greedily to a (1-1/e) approximation factor. They assume the given graph is separated into a small number of communities to apply their fairness notion to. While this also admits a greedy approach via Monte Carlo methods, it requires too long to run.

Their main contribution is a (1-1/e-epsilon) approximate and efficient solution to IM under welfare fairness. En route, they provide an unbiased estimator of the fair objective under a number of random seeds for which edges spread influence when. This is harder than the vanilla problem because the objective is raised to a power alpha in [0,1] as opposed to just 1. They show how to use a Taylor series to provide an unbiased estimator. This is used to then compute the marginal expected gain in influence of adding a vertex to their set and thus greedily select the best one.

They also experimentally validate their results on five different, semi-synthetic data (e.g., for their Flixster data, they had to construct community information). They show that the price of fairness and effect of fairness both decrease as the alpha parameter increases, which makes sense since that decreases the fairness impact on the objective. They also show that adding more influence budget generally decreases the price and effect of fairness.

**Strengths:**

To my knowledge, the theoretical results of this paper are substantial and original. It’s always interesting and practical to see how notions of fairness apply to different problems. They are the first to efficiently solve IM under welfare fairness, and their proposal of an unbiased estimator for the marginal influence spread of a vertex seems notable, though I am not particularly familiar with influence literature (e.g., it is possible these methods are pretty standard for influence spreading strategies). From my standpoint, it is a nice result with a solid place in fairness literature.

**Weaknesses:**

While I think the paper is reasonably clear, it can be quite dense at times with its heavy use of notation. I found it particularly hard to follow in Section 3.4, when they pretty much state two lemmas and a theorem, deferring most proofs to the appendix. I think it’s better to explain these things at a high level more and defer formal statements to the end. It was hard to make much of it. In general, I would prefer the paper to undergo more edits for clarity, particularly in Section 3.4, before publication.


**Questions:**

1. A big value of your work is that your algorithm is efficient. Do you have any experiments that evaluate the efficiency?
2. Can you give a high level discussion of the results from Section 3.4?
3. For someone not in the area of influence spreading, can you discuss how natural the welfare fairness metric is? It's somewhat unintuitive at a glance.
4. Big question: what is the runtime of your algorithm?

**Limitations:**

I do not see a limitations section. Most papers have limitations of one kind or another, even highly theoretical ones like this. However, I don’t think it is the most necessary in this case, as no subjects are involved and they are only suggesting a general tool.

---

> ### Author Rebuttal · Authors · 2023-08-05
>
> ### **Response to Reviewer RYE8**
>
> Thank you for your insightful comments and suggestions. We reply to all your comments below.
>
> > **Weakness 1**: While I think the paper is reasonably clear, it can be quite dense at times with its heavy use of notation. I found it particularly hard to follow in Section 3.4, when they pretty much state two lemmas and a theorem, deferring most proofs to the appendix. I think it’s better to explain these things at a high level more and defer formal statements to the end. It was hard to make much of it. In general, I would prefer the paper to undergo more edits for clarity, particularly in Section 3.4, before publication.\
> >**Question 2**: Can you give a high level discussion of the results from Section 3.4?
>
> **A1**: Thanks for your suggestion. The main idea of Section 3.4 is to analyze how many RR sets we need to generate, such that the proposed algorithm could have a good lower bound with theoretical guarantee.
>
> To be exact, we find that when we generate $\theta= \frac{12Q(\log{2}+\log{CQ}+\ell\log{n})}{\varepsilon^2(1-b_0)}\geq max(\theta_1, \theta_2)$ RR sets, our algorithm can guarantee a $1-1/e-\varepsilon$ approximation towards the optimal solution with a probability at least $1-1/n^\ell$ for any $\varepsilon>0$, $\ell>0$, and $Q\geq 2$.
>
> Moreover, we have summarized important symbols in our paper and presented them in the following Table:
>
> | Symbol | Explanation |
> |--|--|
> | $G=(V, E, p)$ | network |
> | $V$ | node set of the network |
> | $E$ | edge set of the network |
> | $n_G$ | number of nodes in $G$, i.e. $n_G=\|V\|$ |
> | $p(u,v)$ | probability that $u$ influence $v$ |
> | $S$ | seed set |
> | $S^*$ | the optimal seed set for fair influence maximization |
> | $ap(v,S)$ | the expected probability that $v$ is activated by $S$ |
> | $\sigma(S)$  | influence spread of $S$, i.e. $\sigma(S) = \sum_{v\in V} ap(v,S)$ |
> | $\mathcal{C}=\\{c_1,c_2,c_3,\cdots\\}$  | community structure |
> | $V_c$ | node set in community $c$ |
> | $n_c$ | number of nodes in community $c$, i.e. $n_c = \|V_c\|$ |
> | $\bf{u}_c$ | utility of $c$ (expected fraction of influenced nodes in $c$) |
> | $F_\alpha(S)$ | fair influence of $S$ |
> | $\mathcal{R}$ | the set of RR sets |
> | $\mathcal{R}_c$ | the set of RR sets rooted in community $c$ |
> | $\hat{F}_\alpha(S,\mathcal{R})$ | the unbiased estimator for fair influence of $S$ based on $\mathcal{R}$|
> | $\theta$ | the number of RR sets |
> | $\theta_c$ | the number of RR sets rooted in community $c$ |
> | $\alpha$ | aversion parameter regarding fairness |
> | $Q$ | approximation parameter for Taylor expansion |
> | $\varepsilon$ | accuracy parameter |
> | $\ell$ | confidence parameter |
>
>
> >**Question 1**: A big value of your work is that your algorithm is efficient. Do you have any experiments that evaluate the efficiency?\
> >**Question 4**: Big question: what is the runtime of your algorithm?
>
> **A2**: When the number of communities is small (e.g., a constant), then the time complexity of Algorithm 2 in our paper mainly depends on the number of generated RR sets $\theta$. Furthermore, the value of $\theta$ is mainly determined by both the size of the network and the number of communities.
>
> In the following, we exhibit the runtime of our algorithm with the scale of networks where we set $\alpha=0.5$, $k=50$, $\varepsilon=0.1$, $\ell=1$, and $Q=2$. Note that our algorithm is currently implemented in **Matlab**, thus it costs more time to generate RR sets (generating RR sets in C++ could be about 100 times faster). In the following Table, we test four networks with ground-truth community structure where **RR sets** refers to the time (seconds) used to generate RR sets, **IMM** and **FIMM** denote the time used to select seeds based on the generated RR sets for IMM and FIMM, respectively.
>
> |  Network | $n$ | $C$ | RR sets | IMM | FIMM |
> | --: | --: | --: | --: | --: | --: |
> | Email | 1,005 | 42 | 14.281  | 0.011 | 0.020 |
> | Amazon | 12,698 | 509 | 82.204 | 0.028 | 0.283 |
> | Youtube | 30,696 | 1,157 | 162.533  | 0.052  | 3.592  |
> | DBLP | 72,875 | 1,352 | 735.755  | 0.063 | 18.259 |
>
>
> >**Question 3**: For someone not in the area of influence spreading, can you discuss how natural the welfare fairness metric is? It's somewhat unintuitive at a glance.
>
> **A3**: Assuming we are trying to instruct a notice over a crowd consisting of three disjoint communities of the same size by selecting some influencers to spread the notice, the first solution leads to the influenced fraction in each community as 0.16, 0.16, and 0.81, and the second solution leads to that as 0.36, 0.36, and 0.36. Influence maximization would prefer the first solution since the total influenced fraction is 1.12, bigger than 1.08 as in the second solution. However, influence maximization with welfare fairness ($\alpha=0.5$) would prefer the second solution since $0.36^{0.5}+0.36^{0.5}+0.36^{0.5} = 1.8$, bigger than $0.16^{0.5}+0.16^{0.5}+0.81^{0.5}=1.7$ as in the first solution.
>
> The natural welfare fairness metric can also be described by the **Transfer Principle** mentioned in [r7] (ref [7] in our paper). Consider individuals $i$ and $j$ in utility vector $\bf{u}$ such that $u_i<u_j$. Let $\bf{u'}$ be another utility vector that is identical to $\bf{u}$ in all elements except $i$ and $j$ where $u'_i = u_i + \delta$ and $u'_j = u_j - \delta$ for some $\delta \in (0, (u_j-u_i)/2)$. Then, $W(\bf{u}) < W(\bf{u'})$. Informally, transferring utility from a high-utility to a low-utility individual should increase social welfare.
>
> In general, welfare fairness asks to find a solution that could lead to a fairer allocation (smaller gap) among different groups, with a limited sacrifice in the total influence.
>
> [r7] Aida Rahmattalabi, Shahin Jabbari, Himabindu Lakkaraju, Phebe Vayanos, Max Izenberg, Ryan Brown, Eric Rice, and Milind Tambe. Fair influence maximization: A welfare optimization approach. In AAAI, volume 35, pages 11630–11638, 2021.

---

> ### Author Response · Authors · 2023-08-20
> **Thanks for your comments**
>
> Dear Reviewer RYE8,
>
> We would like to express our sincere gratitude to you for reviewing our paper and providing valuable feedback. Could we kindly know if our responses have addressed your concerns? If there are any further questions, we are happy to clarify. Thank you.
>
> Best,
>
> All authors

---

### Official Review · Reviewer_ScbK · 2023-07-07

**Soundness:** 3 good
**Presentation:** 2 fair
**Contribution:** 3 good
**Rating:** 6
**Confidence:** 4

**Summary:**

The authors study a variant of the popular influence maximization problem
that incorporates fairness constraints. Given a partition of the node set
of the graph into communities, the goal is to select seeds subject to
budget constraints that maximizes influence as well as reduces the
influence gap between communities. The authors consider a particular notion
of fairness called welfare fairness, where a single parameterized objective
function can be used to achieve the desired balance between maximizing
influence and maintaining fairness. The objective function is monotone
submodular, and therefore, the greedy algorithm guarantees a constant
approximation factor as in the case of the traditional influence
maximization problem. Since the greedy algorithm is not scalable to large
networks, this paper focuses on developing an efficient algorithm using the
reverse influence sampling (RIS) approach. The main contributions are an
unbiased estimator for the fractional power of the arithmetic mean using
Taylor expansion, solving a main hurdle for using RIS approach for welfare
fairness objective function, and an analysis of the greedy algorithm based
on the RIS approach. Experiments are carried out on five social networks to
study the trade-off between fairness and influence spread.

**Strengths:**

The paper addresses an important topic of scalability in influence
maximization with fairness. The unbiased estimator for the fractional power
of the arithmetic mean is a significant contribution of this paper, which
could be applicable to other problems as well. The paper addresses
important computational and approximation related aspects.

The paper is generally well written.

**Weaknesses:**

The importance of/need for the unbiased estimator is not discussed.
Firstly, in Section 3.1, it is pointed out that (\hat{u}_c)^\alpha is a
biased estimator of u_c^\alpha. Suppose we were to use it, how bad would
the outcome be?

A second related issue is that the paper does not comment on the complexity
of computing the unbiased estimate. When the number of RR sets is large,
updating according to equation (5) can be compute intensive (Algorithm 2
line 11).

Also, some approximation of equation (5) must be involved as the summation
over n goes from 1 to infinity. The authors do not seem to discuss about
the truncation of the Taylor series and its effect on the accuracy.

In Theorem 1, the role of \epsilon is not clear. It does not seem to
feature in the probability expression or have anything to do with the size
of the RR set. In Lemma 4, it is mentioned, but again, its role is not at
clear.

Experiments section could be much stronger: Considering that the key problem
addressed by the paper is scalability, the experiments section does
not provide any analysis of the computation time of the algorithm and how
it scales with the size of the networks. It also does not provide any
analysis regarding how truncation of higher order terms in the Taylor
series affect the performance. Thirdly, there is no analysis of how it
scales with the size of the RR set.

Minor comments:
Line 82: "However, these notions can hardly ..." It would be better if this
sentence was supported by references.

Line 197: that covered ... -> that are covered ...

Line 219: then it holds ... -> then the following holds: ...

Line 231: ... holds at least ... -> holds with probability at least ...

**Questions:**

There are some questions in the Weaknesses that need to be addressed.

In algorithm 1, line 6, is it sample with replacement?

In algorithm 1, line 9, what if c(v) is same as c(u)?

In algorithm 2, line 20, what is \eta(v)?

What is \Theta in Lemma 3? It is not defined anywhere.

**Limitations:**

Limitations not adequately addressed.

No negative societal impact.

---

> ### Author Rebuttal · Authors · 2023-08-04
>
> ### **Response to Reviewer ScbK**
>
> Thank you for your thorough and insightful comments as well as acknowledging our theoretical analysis. We reply to all the points below.
>
> >**Weakness 1**: The importance of/need for the unbiased estimator is not discussed. Firstly, in Section 3.1, it is pointed out that $(\hat{u}_c)^\alpha$ is a biased estimator of $u_c^\alpha$. Suppose we were to use it, how bad would the outcome be?
>
> **A1**: Thank you for addressing this interesting problem. As indicated by [r27] (ref [27] in our paper) which proposes an unbiased estimator for powers of the arithmetic mean, "In any problem for which an unbiased estimator is thought desirable, the proposed unbiased estimator $\hat{x}^m$ ($m=2,3,\cdots$) is naturally preferable, although **the bias in $\hat{x}^m$ may not be very large**. Furthermore, for many, if not most, of the populations that may be expected to arise in statistical work **the variance of $\hat{x}^m$ will be smaller than the mean square of$\bar{x}^m$**".
>
> In our paper, we propose an unbiased estimator for the fractional powers of the arithmetic mean based on Taylor expansion and the unbiased estimator for the integer powers of the arithmetic mean. Therefore, we can calculate both the variance of our proposed unbiased estimator and the mean square error of the biased $(\hat{u}_c)^\alpha$ following the analysis in section 3 of [r27].
>
> Moreover, though the outcome may not be too bad when using the biased estimator, our theoretical analysis cannot hold based on such a biased estimator and we will not able to give an lower-bound guarantee on our algorithm.
>
> [r27] Gerald J Glasser. An unbiased estimator for powers of the arithmetic mean. Journal of the Royal Statistical Society: Series B (Methodological), 23(1):154–159, 1961.
>
> >**Weakness 2**: A second related issue is that the paper does not comment on the complexity of computing the unbiased estimate. When the number of RR sets is large, updating according to equation (5) can be compute intensive (Algorithm 2 line 11).
>
> **A2**: As you mentioned, updating according to Eq.(5) can be computationally intensive when we compute the summation over $n$ from 1 to infnity. However, in experiments, we only keep the first two terms, which greatly reduces the computation cost. The reason that we truncate the Taylor series to the first two terms is detailed in the next response **A3**.
>
> >**Weakness 3**: Also, some approximation of equation (5) must be involved as the summation over n goes from 1 to infinity. The authors do not seem to discuss about the truncation of the Taylor series and its effect on the accuracy.
>
> **A3**: We were sorry that we did not make it clear in our main paper. Actually, the $Q$ (referred to as the approximation parameter) in Lemma 3 and Lemma 4 indicates the truncation of the Taylor series where we only keep the first $Q$-terms. Moreover, Theorem 1 shows that our algorithm is theoretically guaranteed for any $Q\geq 2$.
>
> >**Weakness 4**: In Theorem 1, the role of \epsilon is not clear. It does not seem to feature in the probability expression or have anything to do with the size of the RR set. In Lemma 4, it is mentioned, but again, its role is not at clear.
>
> **A4**: In general, $\varepsilon$ is a tunable parameter that controls the approximation ratio ($1-1/e-\varepsilon$) between the output of our algorithm and the optimal result. By setting $\varepsilon$ (controls accuracy) and $\ell$ (controls confidence), which are two standard parameters in IMM, we can determine the number of RR sets need. The smaller the $\varepsilon$ is, the higher the approximation ratio can achieve, and the more RR sets should be generated. Besides, following the default setting of IMM, we set $\varepsilon=0.1$ and $\ell=1$ in our experiments.
>
> >**Weakness 5**: Experiments section could be much stronger: ......
>
> **A5**: The number of RR sets is mainly determined by both the size of the network and the number of communities. In the following, we exhibit the runtime of our algorithm with the scale of networks. Note that our algorithm is currently implemented in **Matlab**, thus it costs more time to generate RR sets (generating RR sets in **C++** could be more than 100 times faster). **RR sets** refers to the time (seconds) used to generate RR sets, **IMM** and **FIMM** denote the time used to select seeds based on the generated RR sets for IMM and FIMM, respectively.
>
> |Network|$n$|$C$|RRsets|IMM|FIMM|
> |--|--|--|--|--|--|
> |Email|1,005|42|14.281|0.011|0.020|
> |Amazon|12,698|509|82.204|0.028|0.283|
> |Youtube|30,696|1,157|162.533|0.052|3.592|
> |DBLP|72,875|1,352|735.755|0.063|18.259|
>
> >**Question 1**: In algorithm 1, line 6, is it sample with replacement?
>
> **A6**: Yes, it is randomly sampled with replacement, which means each node could be the root for multiple RR sets.
>
> >**Question 2**: In algorithm 1, line 9, what if c(v) is same as c(u)?
>
> **A7**: $\kappa[v][c]$ is an indicator that implies the number of RR sets rooted in the community $c$ that is covered by $v$. Therefore, it still counts when $c(v)=c(u)$ in algorithm 1, line 9.
>
> >**Question 3**: In algorithm 2, line 20, what is \eta(v)?
>
> **A8**: $\eta(v)$ is a linked list from nodes to their covered RR sets. It is initialized when generating RR sets in algorithm 2, line 1, as an output from algorithm 1 (line 11).
>
> >**Question 4**: What is \Theta in Lemma 3? It is not defined anywhere.
>
> **A9**: The $\theta$ in Lemma 3 (and also Lemma 4) indicates the total number of RR sets. It is first introduced in line 187 of our paper.
>
> Besides, we are sorry for our carelessness in forgetting to mention that $\theta = max( \theta_1, \theta_2)$ in Theorem 1.
>
> >**Limitations**: Limitations not adequately addressed.
>
> **A10**: The limitations of our work are mainly two points. The first limitation is that our algorithm can only be efficient when the number of communities is small (e.g., a constant). The second limitation is that our algorithm only works for networks with disjoint groups.

---

> > ### Comment · Reviewer_ScbK · 2023-08-18
> > **Questions about Theorem 1**
> >
> > I thank the authors for carefully answering most of my questions in a satisfactory manner. I had two questions based on the response.
> >
> > 1. Shouldn't Theorem 1 have $\theta$ in the statement? All the guarantees hold only if you have adequate number of RR sets. $\theta$ should appear in Theorem 1 as it appears in the Lemmas.
> > 2. In Lemmas 3 and 4, it seems like as $Q$ is increased, the number of RR sets required also increases. What is the intuition behind it? Smaller the $Q$, the less accurate the estimation. Wouldn't this mean that we need more RR sets to compensate for error in estimation?

---

> > > ### Author Response · Authors · 2023-08-20
> > > **Further Response to Reviewer ScbK about Theorem 1**
> > >
> > > Thanks for your response and insightful comments. We reply to your new concerns below.
> > >
> > > > **Q1**: Shouldn't Theorem 1 have $\theta$ in the statement? All the guarantees hold only if you have adequate number of RR sets. $\theta$ should appear in Theorem 1 as it appears in the Lemmas.
> > >
> > > **A1**: As we stated in our previous rebuttal **A9**, we are terribly sorry for our carelessness in forgetting to claim $\theta=max(\theta_1, \theta_2)$ in Theorem 1. Just as you mentioned, all the guarantees hold only if we have an adequate number of RR sets. In fact, we have already revised Theorem 1 in the local version of our paper as follows:
> > >
> > > **Theorem 1**: For every $\varepsilon > 0$, $\ell > 0$, $0<\alpha<1$, and $Q\geq2$, by setting $\theta\geq max(\theta_1, \theta_2)$, the output $S$ of **FIMM** satisfies $F_\alpha(S) \geq \left(1 - 1/e - \varepsilon\right) F_\alpha(S^*)$, where $S^*$ denotes the optimal solution with probability at least $1-1/n^\ell$.
> > >
> > >
> > > > **Q2**: In Lemmas 3 and 4, it seems like as $Q$ is increased, the number of RR sets required also increases. What is the intuition behind it? Smaller the $Q$, the less accurate the estimation. Wouldn't this mean that we need more RR sets to compensate for error in estimation?
> > >
> > > **A2**: Thanks for raising this interesting question. We have found that the reason lies behind the proof of Lemma 3 and Lemma 4. In short, when we prove $Pr\\{\sum_{i=1}^Q A_i < \sum_{i=1}^Q B_i \\}$ is small enough, we transfer it to an upper bound $1 - \prod_{i=1}^Q(1- Pr\\{A_i < B_i \\}) \leq \sum_{i=1}^Q  Pr\\{A_i < B_i \\}$ and then prove this upper bound is small enough. Therefore, as $Q$ increases, the number of the summed terms increases, leading to a higher upper bound and the need for more RR sets.
> > >
> > > In summary, as $Q$ increases, we need more RR sets to **guarantee the accurate estimation of each term** ($i=1,2,...,Q$). However, the accuracy of the whole estimation is controlled by $\varepsilon$ and holds approximation $1-1/e-\varepsilon$ with probability at least $1-1/n^\ell$, as long as $Q \geq 2$.
> > >
> > > In the future, we will try to explore a more optimal way (e,g, finding a tighter upper bound) to prove Lemma 3 and Lemma 4 with fewer RR sets.

---

> > > > ### Comment · Reviewer_ScbK · 2023-08-20
> > > >
> > > > A1: Got it. But the statement is not quite right in the current form. Instead, something like the following might be better:
> > > >
> > > > Theorem 1: For every $\varepsilon > 0$, $\ell > 0$, $0<\alpha<1$, and $Q\geq2$,  there exists a positive integer $\theta$, such that the output $S$ of FIMM with $\theta$ RR sets satisfies $F_\alpha(S) \geq \left(1 - 1/e - \varepsilon\right) F_\alpha(\hat{S})$, where $\hat{S}$ denotes the optimal solution with probability at least $1-1/n^\ell$.
> > > >
> > > > In the previous version, it is assumed that $\theta_1$ and $\theta_2$ are known. However, those two constants appear only in the proof of the theorem.
> > > >
> > > > A2: It is a reasonable explanation. Thanks.

---

> > > > > ### Author Response · Authors · 2023-08-20
> > > > > **Thanks for your response and suggestion**
> > > > >
> > > > > Thanks again for your insightful comment! It has been a great pleasure discussing the paper with you.
> > > > >
> > > > > As you mentioned, it would be better to claim there exists a positive integer $\theta$ rather than claim $\theta \geq (\theta_1, \theta_2)$. We will revise Theorem 1 as you suggested.
> > > > >
> > > > > Moreover, If there are any further questions, we are happy to clarify. Thank you.
> > > > >
> > > > >
> > > > >
> > > > > Best,
> > > > >
> > > > > All authors

---

### Official Review · Reviewer_tPHQ · 2023-07-08

**Soundness:** 3 good
**Presentation:** 2 fair
**Contribution:** 2 fair
**Rating:** 3
**Confidence:** 4

**Summary:**

This paper aims to study the problem of fairness-aware influence maximization over a community structure under a welfare fairness notion that balances fairness level and influence spread using an exponentially-weighted sum over communities objective with an fractional exponent parameter α expressing inequality aversion. Given an unbiased estimator for the fractional powers from [27], a reverse-influence-sampling approach, adapted from previous works, is applied to the problem that turns the problem into a weighted maximum coverage problem. The number of reverse reachable (RR) samples needed to achieve a good approximation and an algorithm is given that achieves a 1 − 1/e − ε approximation.

**Strengths:**

S1. Revisiting an important problem with an aim to efficiency.
S2. Lucrative application of reverse influence sampling approach.
S3. Approximation guarantees.

**Weaknesses:**

W1. Limited novelty with respect to previous work in [7].
W2. Lack of discussion of the most basic problem case of full fairness.
W3. Lack of discussion of and comparison to most recent related work.

**Questions:**

The solution relies on a submodularity of the objective function and the accompanying analysis in [7]. However, there is no discussion of what happens in the most interesting case from the point of view of fairness, namely as α tends to 0. According to previous works, the objective is not submodular in exactly that case. This matter deserves discussion yet it is not addressed. The work pays no attention to more recents works in the area, such as [A, B]. There is no experimental comparison to the algorithms developed in those works, which, at least the ones in [A], also claim efficiency, and no discussion in terms of the framework presented in [B].

[A] Fish et al.: Gaps in information access in social networks? WWW 2019.
[B] Farnadi et al.: A unifying framework for fairness-aware influence maximization. Companion of The Web Conference 2020.

**Limitations:**

The paper offers a discussion on the ethical side of the objective aimed for.

---

> ### Author Rebuttal · Authors · 2023-08-05
>
> ### **Response to Reviewer tPHQ**
>
> We thank the reviewer for the insightful comments and suggestions. We reply to all your comments below.
>
> >**Weakness 1**: Limited novelty with respect to previous work in [7].
>
> **A1**: Literature [7] proposes a novel notion of fairness for influence maximization based on cardinal social welfare theory but gives no algorithm. In the field of influence maximization, designing efficient algorithms is also of high significance. Our work just adopts the objective function of cardinal welfare fairness proposed in [7] and **focuses on designing an efficient algorithm to maximize that objective function**. Therefore, the main contribution and key idea of our work are fundamentally different from those in [7].
>
>
> >**Weakness 2**: Lack of discussion of the most basic problem case of full fairness.\
> >**Question 1**: The solution relies on a submodularity of the objective function and the accompanying analysis in [7]. However, there is no discussion of what happens in the most interesting case from the point of view of fairness, namely as $\alpha$ tends to 0. According to previous works, the objective is not submodular in exactly that case. This matter deserves discussion yet it is not addressed.
>
> **A2**: According to [7], as long as $\alpha<1$ and $\alpha\neq 0$, the objective function $W_\alpha(u) = \sum_{c \in C} \textbf{u}_c(S)^\alpha$ satisfy 5 of their mentioned principles. Then, based on Lemma 1 in [7], any welfare function that satisfies their 5 principles is monotone and submodular in the influence maximization problem. Therefore, **the objective function adopted in our paper is still submodular in the influence maximization problem even as $\alpha$ tends to 0**.
>
> In addition, as you mentioned, the case of full fairness ($\alpha$ tends to 0) is quite interesting. Under such a setting, any community $c$ that has $\textbf{u}_c(S)>0$ would have a fair utility $\textbf{u}_c(S)^\alpha$ tends to 1. The problem would become a reachable problem that seeks to select a seed set $S$ that could reach the most number of communities. However, since $\alpha>0$, it still satisfies submodularity, thus we did not further discuss it in our paper.
>
>
> >**Weakness 3**: Lack of discussion of and comparison to most recent related work.\
> >**Question 2**:  The work pays no attention to more recent works in the area, such as [A, B]. There is no experimental comparison to the algorithms developed in those works, which, at least the ones in [A], also claim efficiency, and no discussion in terms of the framework presented in [B].
>
> **A3**: As you mentioned, [A] and [B] are related to our work to some extent. However, due to the page limit, we did not to discuss them in the submitted version of our paper, the reasons of which are elaborated in the following.
>
> [A] proposes the $\phi$-MEAN fairness where $\mu_\phi(S,V)=(\frac{1}{|V|} \sum_{v\in V} p_v^\phi)^{1/\phi}$. They study the access gap over bipartitions and $k$-imbalance under different $\phi$. In contrast, our paper addresses the problem of designing efficient algorithms with theoretical guarantees based on existing notions of fairness. Besides, [A] find that only $\mu_{-\infty}$ is balanced which corresponds to the **maximin fairness (MMF), which is also the only setting they test in experiments**. Therefore, we did not further discuss [A] in our paper as we already introduced MMF.
>
> [B] is an interesting work that develops a formalism which is general enough for specifying different notions of fairness. Their framework covers 4 fair notions, i.e., Equality, Equity, Maximin (MMF), and Diversity (DC). **These 4 fair notions are already introduced in our Related Work**. Our paper adopts the objective function of welfare fairness, which can better control the trade-off between fairness and total influence. Therefore, we did not further discuss [B] in our paper.
>
> Besides, for the most recent related work, such as MMF (the same in [A]) and DC, the comparison between those fair notions and welfare fairness (which is adopted in our paper) is already presented in [7]. The key idea of our work is to design an efficient algorithm for fair influence maximization based on welfare fairness. Datasets used in [B] contain only 5,00 nodes while our method could handle networks with more than 70,000 nodes efficiently.
>
> Moreover, regarding efficiency, [A] actually claims that **“since our goal in this paper is to introduce and validate a method for reducing access gaps, we will not focus on achieving the fastest implementations”**. In experiments, they show that their method is **"prohibitively slow"** with Greedy (costs more than 20 minutes on a network with only 1,294 nodes). Their method can only be efficient when adopting heuristics (such as Random, Gonzalez, and Naïve Myopic) with poor approximation ratios.

---

> > ### Comment · Reviewer_tPHQ · 2023-08-18
> > **A comparison to the work of Fish et al. is due.**
> >
> > Thank you for your response.
> > The fact that MMF is introduced in this paper does not constitute a valid reason to not discuss and not compare to [A].
> > On the contrary, is a comparison is due exactly for this reason.

---

> > > ### Author Response · Authors · 2023-08-18
> > > **Further Response to Reviewer tPHQ**
> > >
> > > Thanks for your response and constructive comments. Since the submitted version of the paper is not allowed to revise at the current stage, we focused on explaining our ideas behind the texts in our previous rebuttal.
> > >
> > > **We agree with you that it would be better to discuss [A] in our paper**. Actually, we have already added such content in our local version as follows:
> > >
> > > Based on the cardinal welfare theory, the objective function of welfare fairness is to maximize weighted summation over the exponential influenced fraction of all communities. Fish et al. [A] also follow welfare functions and propose $\phi$-mean fairness, where the objective function becomes MMF when $\phi=-\infty$.

---

> > > ### Author Response · Authors · 2023-08-20
> > > **Thanks for your comments**
> > >
> > > Dear Reviewer tPHQ,
> > >
> > > We would like to express our sincere gratitude to you again for reviewing our paper and providing insightful comments. We have now discussed both [A] and [B] in our paper. Besides, Could we kindly know if our previous responses have addressed your concerns? We are happy to clarify if there are any further questions. Thank you.
> > >
> > > Best,
> > >
> > > All authors

---

### Author Response · Authors · 2023-08-17
**We look forward to the reviewers' reply**

We thank all the reviewers again for their insightful comments and constructive suggestions. We valued all the feedback and made a great effort in writing our responses.

We want to kindly draw your attention that it has been a week since we submitted our responses, would you mind letting us know if our responses address your concerns? If you find there are other issues, please kindly let us know, we are happy to follow up with you before the discussion phase ends.

---

> ### Comment · Area_Chair_j8eB · 2023-08-18
>
> Dear authors,
>
> Thank you for your rebuttal, and your messages.
>
> The reviewers and myself will take all of your points into consideration when making our decision.
>
> Best regards.

---

### Decision · Program_Chairs · 2023-09-21

**Decision:**

Accept (poster)

**Comment:**

This paper studies a fair influence maximization problem on large-scale graphs. Three reviewers holds a weakly positive view of the paper. A fourth reviewer has, instead, an unfavorable view of the paper.
Given the reviews, the rebuttal, the subsequent discussions, and my own assessment of the paper, I concur with the positive reviewers --- I believe that this paper would be a nice addition to the NeurIPS program.